# Contextually Guided Transformers via Low-Rank Adaptation

## Abstract

Large Language Models (LLMs) based on Transformers excel at text processing, but their reliance on prompts for specialized behavior introduces computational overhead. We propose a modification to a Transformer architecture that eliminates the need for explicit prompts by learning to encode context into the model's weights. Our Contextually Guided Transformer (CGT) model maintains a contextual summary at each sequence position, allowing it to update the weights on the fly based on the preceding context. This approach enables the model to self-specialize, effectively creating a tailored model for processing information following a given prefix. We demonstrate the effectiveness of our method on synthetic in-context learning tasks and language modeling benchmarks. Furthermore, we introduce techniques for enhancing the interpretability of the learned contextual representations, drawing connections to Variational Autoencoders and promoting smoother, more consistent context encoding. This work offers a novel direction for efficient and adaptable language modeling by integrating context directly into the model's architecture.

## 1 Introduction

Transformer models, laying at the foundation of many modern Large Language Models (LLMs), are exceptionally powerful at understanding and generating text. One of the efficient ways for guiding and specializing their behavior is through the use of prompts – instructions or examples provided at the beginning of an input sequence. These prompts steer model's attention guiding its output towards a desired specialized behavior. However, there's a trade-off. Prompts, especially lengthy or complex ones, increase the amount of data the model has to process during inference running with the same prompt again and again. This additional processing translates into higher computational costs and larger latencies thus motivating an exploration of alternative approaches. An active research direction (Phang et al., 2023) is to adjust the model's internal parameters $\theta \to \theta + \delta\theta(\rho)$ to replicate the outcome of applying a specific prompt $\rho$. This approach effectively incorporates the desired behavior directly into the model. Consequently, the prompt becomes unnecessary during inference, resulting in faster and more resource-efficient computations. Existing methods for transforming prompts into weight updates (Phang et al., 2023) have a couple of common characteristics. First, they tend to treat the prompt as distinct from the main input, handling it separately. Second, they often employ a secondary independently-trained model specifically for interpreting the prompt and calculating the appropriate weight update for the primary model responsible for core language tasks. This separation allows for specialized handling of prompts, but can introduce additional complexity.

In this paper, we propose a novel Transformer design that we call a Contextually Guided Transformer (CGT) that combines both of these components[1] into a single model. For any given Transformer layer $\ell$, our model simultaneously performs two key functions: (a) parses the the input sequence, producing for each token an embedding vector $\boldsymbol{y}^\ell$ that reflects the contextual information (computed at layer $\ell$), and (b) uses this context summary $\boldsymbol{y}^\ell$ to modulate the computation of subsequent layers (beyond layer $\ell$) by effectively generating the weights for these layers. Unlike other approaches, CGT model maintains a summary of the preceding context at each position within the sequence. We train the model to isolate a sequence prefix of any length and compute its corresponding context embedding $\boldsymbol{y}^\ell$. We then freeze this embedding $\boldsymbol{y}^\ell$, along with the generated weights of layers beyond

---

[1] operating on the prompt and operating on the rest of the sequence

Figure 1: **(a)** Model architecture showing processing of a single token with the activation component $\boldsymbol{y}^\nu$ being a function of $(\boldsymbol{x}^{\nu-1}, \boldsymbol{y}^{\nu-1})$ and $\boldsymbol{x}^\nu$ being a function of $\boldsymbol{x}^{\nu-1}$ alone for any $\nu \in [2, \ell]$; activations $\boldsymbol{y}^\ell$ are parameterizing transformations $\mathbf{T}^\kappa(\cdot; \boldsymbol{y}^\ell)$ mapping $\boldsymbol{x}^\nu$ to $\tilde{\boldsymbol{x}}^\nu_{\text{op}} = \mathbf{T}^\kappa \boldsymbol{x}^\nu_{\text{op}}$ for $\nu \geq \ell$ (see Sec. 3.1); **(b)** Auxiliary loss incentivizing the Transformer to encode long-range information in $\boldsymbol{y}$: the auxiliary loss is applied to an input sequence truncated at some random token $\boldsymbol{t}_s$. The context embedding $\tilde{\boldsymbol{y}} := \boldsymbol{y}^\ell_s$ is taken from the Transformer running on the original sequence.

$\ell$, for the remainder of the sequence. This process effectively creates a specialized model tailored for processing the specific sequence following the chosen prefix.

We believe that CGT models can be particularly useful for scenarios where adapting the model's behavior based on an initial context is crucial for effective processing of the subsequent information. We empirically study this technique in three setups: (a) linear regression setup, (b) synthetic in-context learning setup, where the model is presented with multiple demonstrations of an arithmetic task with hidden parameters and (c) text datasets including `c4` and `wikipedia`. In all of these examples, we show that the CGT model maps any given prefix into a specialized model that performs well on the remainder of the sequence. For example, in the in-context learning task, we can convert multiple examples of a task presented in-context into a specialized model capable of solving the corresponding task for new inputs.

Our second contribution is a set of techniques for improving interpretability of the context summary $\boldsymbol{y}^\ell$. Studying $\boldsymbol{y}^\ell$ empirically, we observe that it contains information about the prefix it summarizes, but the context summary encoding is not changing gradually from one token to the next, which makes it difficult to interpret it as a consistent context representation. This motivated us to propose a number of techniques that are aimed at improving the properties of $\boldsymbol{y}^\ell$ endowing it with the desired smoothness prior. Starting with a more grounded approach based on interpreting a Transformer as a Variational Autoencoder, we then derive a simpler regularization scheme that improves the properties of the learned representation $\boldsymbol{y}^\ell$ while also having a positive impact on the performance of the underlying model. We believe that these developed techniques could be useful in a more general setting of sequence representation learning. Equipped with the knowledge of the properties of the underlying semantic representation (such as it's characteristic time scale), one could use our proposed VAE model to discover representations adhering to this "smoothness prior".

## 2 RELATED WORK

**Learning representations with various scales.** Publications (Xu et al., 2022; Tang et al., 2022; Rao et al., 2021; Chen et al., 2023) have explored techniques for assessing the significance of individual tokens with varying levels of detail, aiming to reduce computational overhead. Specifically, (Xu et al., 2022) shares a conceptual similarity with our approach, which involves applying distinct update mechanisms to tokens based on their importance. While their approach distinguishes between informative and placeholder tokens, ours divides embedding dimensions into two segments, each tasked with capturing either local or global context. Also, while (Schmidhuber, 1992) employs a dual-network architecture to address temporal dependencies in sequential data, where the first network dynamically adjusts weights for the second to adapt to temporal patterns, and (Mujika et al., 2017) proposes a recurrent neural network (RNN) with heterogeneous cell types to capture both long-term and short-term dependencies, neither approach facilitates on-the-fly weight updates based on contextual information.

**Transformer + VAE.** Integrating Transformer and Variational Autoencoder (VAE) (Kingma & Welling, 2014) has been a subject of numerous endeavors. (Casale et al., 2018) employs Gaussian processes as priors for the latent space, enabling the model to capture intricate data dependencies. Addressing the issue of controllability in narrative generation, (Wang & Wan, 2019; Fang et al., 2021) develop a conditional VAE framework. (Henderson & Fehr, 2023) introduces a model that incorporates nonparametric variational methods to enhance the information bottleneck in Transformers, leading to better capture of latent representations and improved efficiency across various natural language processing tasks. Similarly to the previous work, our approach proposes a VAE-based method with a meticulously designed regularizer, enabling more flexible control over the representations, ensuring they evolve slowly.

**In-context learning.** In-context learning has garnered significant attention among researchers, particularly with the rise of large language models, owing to its adaptability to unforeseen tasks (Brown et al., 2020). Several studies (Von Oswald et al., 2023; von Oswald et al., 2023; Liu et al., 2022; Min et al., 2022; Zoph et al., 2022) have examined the mechanics of in-context learning to grasp its functionality and rationale. Especially, (Hendel et al., 2023) argues that LLMs learn to represent tasks as vectors in their activation space. When presented with in-context examples, the model constructs a task vector that guides its predictions for new inputs. This work highlights the role of representation learning in ICL and suggests that LLMs can effectively capture task-specific information from few-shot examples. However, it is limited to relatively simple tasks like word mappings and further does not adjust the weights, limiting its adaptability to complex tasks. In contrast, our proposed CGT effectively extracts global context from prompts and generates task-specialized models through weight modulations.

## 3 METHOD

In this section, we detail two core components of our method: (a) our model that simultaneously summarizes the context at layer $\ell$ and uses it to generate weights for layers above $\ell$ and (b) techniques for enforcing a *smoothness prior* on the context representation. The outline can be summarized as follows:

1. In Section 3.1, we describe the CGT *model architecture* that simultaneously computes the context representation $\boldsymbol{y}^\ell$ and uses it to modulate local computation above layer $\ell$.

2. In Section 3.2, we detail the *auxiliary loss* $\mathcal{L}_{\mathrm{aux}}$ that we use for making it possible to freeze $\boldsymbol{y}^\ell$ at any point in the sequence effectively generating a model specialized for the remainder of that sequence.

3. Then, in Section 3.3, we discuss the *smoothness prior* on the context representation $\boldsymbol{y}^\ell$.

4. Finally, in Section 3.4, we introduce our *element-wise regularization technique* for incentivizing representation smoothness. A more elegant *VAE-based approach* and the path to deriving the element-wise regularization method are outlined in Appendix A.

In the following, we consider a modified autoregressive causal Transformer model with $L$ layers, where each layer is represented by embeddings $\boldsymbol{z}^\nu$ with $\nu = 1, \ldots, L$. The model input is a sequence of $n$ tokens $\boldsymbol{t} := \{\boldsymbol{t}_1, \ldots, \boldsymbol{t}_n\}$ with each token taking value in a finite set of all possible tokens $\mathcal{T}$.

### 3.1 MODEL ARCHITECTURE

A central principle of our architecture shown in Figure 1(a) is a carefully designed separation of information flows, enabling efficient computation of the entire model for a given value of $\boldsymbol{y}^\ell$ (which can be either sequence-dependent or fixed). The model is divided into two stages.

The first stage processes the input sequence and generates a contextual representation $\boldsymbol{y}^\ell$, which summarizes information from all preceding tokens at each position. At each layer $\nu \leq \ell$ our model maintains two independent sets of activations $\boldsymbol{x}^\nu$ and $\boldsymbol{y}^\nu$ produced by two sets of Transformer heads. Crucially, the self-attention and MLP operations within these layers are structured to ensure distinct information pathways: $\boldsymbol{x}$ components are processed based on prior $\boldsymbol{x}$ components alone, while $\boldsymbol{y}$ has a full visibility of both $\boldsymbol{x}$ and $\boldsymbol{y}$. This separation ensures that $\boldsymbol{x}$ remains independent of $\boldsymbol{y}$ at each layer before $\ell$, while the final $\boldsymbol{y}^\ell$ at layer $\ell$ aggregates information from both pathways across all preceding layers. Importantly, this also allows the the value of $\boldsymbol{y}^\ell$ to be provided externally by simply bypassing the $\boldsymbol{y}^\ell$ pathway, thus reducing computational cost.

At the second stage, we use the contextual representation $\boldsymbol{y}^\ell$ to modulate the processing of subsequent layers $\nu > \ell$. For each layer $\nu > \ell$, $\boldsymbol{y}^\ell$ dynamically generates the weights of linear operators $\mathbf{T}^\kappa$, where $\kappa = 1, \ldots, 2(L - \ell)$. These operators are applied before each MLP and self-attention operation (a total of $2(L - \ell)$ operators). The weights of $\mathbf{T}^\kappa(\boldsymbol{y}^\ell)$ are generated independently for each sequence position $s$ based on the corresponding $\boldsymbol{y}_s^\ell$. This mechanism allows the global context summarized in $\boldsymbol{y}^\ell$ to influence the processing of each token. In this stage, the model activations only consist of $\boldsymbol{x}$. The $\boldsymbol{y}^\ell$ representation is fixed and used solely for generating the weights of $\mathbf{T}^\kappa(\boldsymbol{y}^\ell)$, which are applied to the $\boldsymbol{x}$ activations at each layer. Specifically, instead of passing $\boldsymbol{x}$ to an MLP or self-attention layer, we instead pass $\tilde{\boldsymbol{x}} := \mathbf{T}^\kappa(\boldsymbol{y}^\ell)\boldsymbol{x}$.

Our model defines the linear operators $\mathbf{T}^\kappa$ using a low-rank weight generator:

$$\mathbf{T}^\kappa\left(\boldsymbol{x}; \boldsymbol{y}^\ell\right) := \boldsymbol{x} + \delta\mathbf{W}^\kappa\left(\boldsymbol{y}^\ell\right)\boldsymbol{x} \quad \text{with} \quad \delta\mathbf{W}_{ij}^\kappa\left(\boldsymbol{y}^\ell\right) = \sum_{k=1}^{r} \boldsymbol{L}_{ik}^\kappa\left(\boldsymbol{y}^\ell\right)\boldsymbol{R}_{jk}^\kappa\left(\boldsymbol{y}^\ell\right),$$

where $\delta\mathbf{W}^\kappa(\boldsymbol{y}^\ell)$ represents a change in weights applied to $\boldsymbol{x}$, and it is generated using a low-rank decomposition:

$$\boldsymbol{L}^\kappa\left(\boldsymbol{y}^\ell\right) := \sum_{m=1}^{M} \boldsymbol{L}^{\kappa,m}\sigma_m^\kappa\left(\boldsymbol{y}^\ell\right), \quad \boldsymbol{R}^\kappa\left(\boldsymbol{y}^\ell\right) := \sum_{m=1}^{M} \boldsymbol{R}^{\kappa,m}\sigma_m^\kappa\left(\boldsymbol{y}^\ell\right).$$

Here $\boldsymbol{L}^{\kappa,m} \in \mathbb{R}^{(\dim \boldsymbol{x}) \times r}$ and $\boldsymbol{R}^{\kappa,m} \in \mathbb{R}^{(\dim \boldsymbol{x}+1) \times r}$ are additional learned *template* matrices, $M$ is their total number and $r$ is the rank of the generated $\delta\mathbf{W}$. We use $\dim \boldsymbol{x} + 1$ for $\boldsymbol{R}$ to incorporate a bias term within the linear transformation.

The values of $\sigma(\boldsymbol{y}^\ell) = h(\mathbf{S}^\kappa \boldsymbol{y}^\ell) \in \mathbb{R}^M$ act as template mixing coefficients with a non-linearity $h(\cdot)$ given by either a hyperbolic tangent or softmax. $\mathbf{S}^\kappa \in \mathbb{R}^{M \times (\dim \boldsymbol{y}+1)}$ are learned linear transformations that map $\boldsymbol{y}^\ell$ to a specific mixture of templates used to generate a low-rank $\mathbf{T}^\kappa$. We use $\dim \boldsymbol{y} + 1$ to incorporate a bias term within this transformation as well.

While more complex operators could be used for $\mathbf{T}^\kappa(\boldsymbol{y}^\ell)$, we choose linear[2] operators for efficiency. When $\boldsymbol{y}^\ell$ is fixed and position-independent, these linear operators can be effectively "folded" into the subsequent self-attention and MLP operations. This way, when $\boldsymbol{y}^\ell$ if frozen, we can effectively generate a model with embedding size $\dim \boldsymbol{x}$ and fixed position-independent weights that runs on $\boldsymbol{x}$ activations alone.

### 3.2 Auxiliary Loss

Our model was designed to use a self-modulation mechanism controlled by the context representation $\boldsymbol{y}^\ell$. Since $\boldsymbol{y}^\ell$ is generally token-dependent, the transformations $\mathbf{T}^\kappa(\boldsymbol{y}^\ell)$ are also generally evolving from token to token. However, if $\boldsymbol{y}^\ell$ were fixed all $\mathbf{T}^\kappa$ would become token-independent, which would allow us to "fold" these linear maps into the following linear operators.

While it would be natural to expect the *context representation* $\boldsymbol{y}^\ell$ to evolve slowly along the sequence[3], in practice we observe that this property does not generally hold. Furthermore, in few-shot in-context learning tasks, we observe that simply freezing $\boldsymbol{y}^\ell$ does not generally lead to accurate specialized models (see Section 4). And since this desired property is not emergent, we need to purposely design training objectives to achieve it.

Our primarily loss function is a conventional cross-entropy loss $\mathcal{L}_{\text{ce}}$, which measures the difference between the predicted probabilities of our modified causal autoregressive Transformer model and the groundtruth shifted input sequence. However, since we also expect our model with frozen $\boldsymbol{y}^\ell$ to perform well, our full loss

$$\mathcal{L} = \eta\mathcal{L}_{\text{ce}} + (1 - \eta)\mathcal{L}_{\text{aux}}$$

includes an additional auxiliary loss $\mathcal{L}_{\text{aux}}$ component with $0 \leq \eta \leq 1$ interpolating between two losses. This auxiliary loss $\mathcal{L}_{\text{aux}}$ is computed as follows (see Figure 1(b)): **(i)** first, we randomly sample a sequence position $s \in (2, n)$ with $n$ being the sequence length; **(ii)** then treating the input sequence prefix $\{\boldsymbol{t}_1, \ldots, \boldsymbol{t}_{s-1}\}$ as our current prompt, we compute the value of $\boldsymbol{y}_{s-1}^\ell$ at the end of it;

---

[2]The way $\mathbf{T}^\kappa(\boldsymbol{y}^\ell)$ acts on $\boldsymbol{x}$ is linear, not the way it's coefficients depend on $\boldsymbol{y}^\ell$.

[3]Assuming that the information about the context is changing incrementally

**(iii)** we build an auxiliary Transformer model with the same weights and a fixed value of $\tilde{\boldsymbol{y}}^\ell = \boldsymbol{y}_{s-1}^\ell$ and apply it to the remainder of the sequence $\{\boldsymbol{t}_s, \ldots, \boldsymbol{t}_n\}$; **(iv)** compute $\mathcal{L}_{\text{aux}}$ as a conventional cross-entropy loss on this subsequence $\{\boldsymbol{t}_s, \ldots\}$.

Our proposed auxiliary loss is designed to force the activation component $\boldsymbol{y}_s^\ell$ (the only component communicating information between two parts of the sequence in the auxiliary loss; see Fig. 1(b)) to summarize the context $\boldsymbol{t}_{\leq s} := \{\boldsymbol{t}_1, \ldots, \boldsymbol{t}_s\}$. Indeed, by optimizing the cross-entropy loss on $\boldsymbol{t}_{>s} := \{\boldsymbol{t}_{s+1}, \ldots, \boldsymbol{t}_n\}$ we maximize the lower bound on the mutual information $\mathbb{I}(\boldsymbol{t}_{>s}; \boldsymbol{y}_s^\ell)$. This in turn reduces the conditional mutual information $\mathbb{I}(\boldsymbol{t}_{>s}; \boldsymbol{t}_{\leq s} | \boldsymbol{y}_s^\ell) = \mathbb{I}(\boldsymbol{t}_{>s}; \boldsymbol{t}_{\leq s}) - \mathbb{I}(\boldsymbol{t}_{>s}; \boldsymbol{y}_s^\ell)$, which can be interpreted as making $\boldsymbol{y}_s^\ell$ condense all the information from the context $\boldsymbol{t}_{\leq s}$ that is useful for generating the rest of the sequence $\boldsymbol{t}_{>s}$.

For improved efficiency with long sequences, we can focus on a more immediate prediction restricted to a fixed horizon $\Delta$. In this case, $s$ is sampled from a range $(2, n - \Delta]$ and the fixed representation $\boldsymbol{y}_{s-1}^\ell$ is applied to a subsequence $\{\boldsymbol{t}_s, \ldots, \boldsymbol{t}_{s+\Delta}\}$ of length $\Delta + 1$. More details about our full model can be found in Appendix B.1.

### 3.3 LEARNING SLOW FEATURES

The interpretation of $\boldsymbol{y}^\ell$ as the context summary intuitively suggests that $\boldsymbol{y}^\ell$ should not change drastically between consecutive tokens. However, in our experiments with the proposed model and the auxiliary loss, we only witnessed the emergence of smooth $\boldsymbol{y}^\ell$ in strongly regularized models (see Section 4). More generally, $\boldsymbol{y}^\ell$ was seen to contain additional irrelevant information and exhibit a non-trivial dependence on the sequence position. Here we propose a set of techniques for enforcing a "slowness prior" on the evolution of $\boldsymbol{y}^\ell$ allowing us to obtain more interpretable context representations. Here we assume that $\boldsymbol{y}^\ell$ viewed as a random variable is a Gaussian process, or in a discretized form, a multivariate Gaussian distribution $p_0(\boldsymbol{y}_1^\ell, \ldots, \boldsymbol{y}_n^\ell)$ with the mean $\langle \boldsymbol{y}_s^\ell \rangle = \boldsymbol{m}_s$ and covariance $\langle (\boldsymbol{y}_s^\ell - \boldsymbol{m}_s)(\boldsymbol{y}_t^\ell - \boldsymbol{m}_t) \rangle$ being given by a known kernel $\mathcal{K}_{s,t}$. If $\mathcal{K}_{s,t}$ decays towards zero with growing $|s - t|$, it ensures that computed at two nearby points in time, the $\boldsymbol{y}^\ell$ activations maintain a certain degree of coherence, but this correlation disappears over time.

In the following, we choose an unbiased prior with $\boldsymbol{m}_s = 0$, but the choice of $\mathcal{K}_{s,t}$ depends on the nature of the generative process. For example, if $\boldsymbol{y}^\ell$ is expected to capture a slowly changing context information with a characteristic temporal scale $\lambda$, the covariance $\mathcal{K}_{s,t}$ would only depend on $|s - t|$ and could scale roughly as $\exp(-|s - t|^2 / 2\lambda^2)$. However, in in-context learning tasks, $\boldsymbol{y}^\ell$ would be expected to change more rapidly at the beginning of the sequence and saturate after enough examples of the task are presented to the model. The corresponding $\mathcal{K}_{s,t}$ would not generally vanish for large enough $s$ and $t$. In Appendix C we consider a trivial example of the sequence mean $\alpha$ estimation with a prior $\alpha \sim \mathcal{N}(0, 1)$ given a sequence of observations $\alpha + \epsilon \beta_s$ with $\beta_s$ sampled iid from $\mathcal{N}(0, 1)$. The covariance matrix for this problem is given by:

$$\mathcal{K}_{s,t} = 1 + \frac{\epsilon^2}{st} \min(s, t). \tag{1}$$

The important characteristic of this covariance matrix is that $\mathcal{K}_{s,t} \to 1$ when both $s$ and $t$ go to infinity. The constant value of $1$ reflects information of the original prior on $\alpha$ and $1/t$ dependence is due to the estimation error disappearing as more and more examples are shown. A proper choice of the prior kernel $\mathcal{K}_{s,t}$ is thus problem-dependent.

### 3.4 ELEMENT-WISE REGULARIZERS

Perhaps one of the most direct ways of incorporating a Gaussian process prior into our model is to view the Transformer as a Variational Autoencoder (VAE). As shown in Appendix A.1, a Transformer model can be viewed as a VAE by replacing conventional $\boldsymbol{y}_s^\ell$ activations with two sets of variables $\boldsymbol{\mu}_s$ and $\boldsymbol{\sigma}_s$ and randomly sampling $y_{i,s}^\ell$ from $\mathcal{N}(\mu_{i,s}, \sigma_{i,s})$. The VAE loss function is then given by:

$$\mathcal{L}_{\text{VAE}}(\boldsymbol{t}) = \mathcal{L}_{\text{rec}} - \frac{\beta_y}{2} \sum_{i,s} \log \sigma_{i,s}(\boldsymbol{t}) + \frac{\beta_y}{2} \sum_{i,s,t} \mathcal{K}_{s,t}^{-1} \mu_{i,s}(\boldsymbol{t}) \mu_{i,t}(\boldsymbol{t}) + \frac{\beta_y}{2} \sum_{i,s} \mathcal{K}_{s,s}^{-1} \sigma_{i,s}(\boldsymbol{t}), \tag{2}$$

where $\mathcal{L}_{\text{rec}}$ is a cross-entropy reconstruction loss.

While being potentially useful across a wide range of sequence representation learning problems, this approach involves stochastic model activations $\boldsymbol{y}^\ell$ and was seen to produce less accurate models

than simpler approaches inspired by it. A similar prior can also be enforced by a moment-based regularization technique (Appendix A.2), but it's reliance on random pairs of sequence elements makes this method computationally expensive and noisy.

Noticing that the $3^{\text{rd}}$ term in the right-hand side of equation 2 can be interpreted as form of continuity regularization (Appendix A.3), we found that an even simpler and less computationally intensive technique produces comparable result. The idea that we used in most of our experiments is to enforce the continuity in $\boldsymbol{y}^\ell$ by simply penalizing large time step differences in $\boldsymbol{y}_i^\ell$, or in its normalized value:

$$\mathcal{R}_C^\ell \sim \sum_{s=2}^{n} \zeta_C(s) \left\| \boldsymbol{n}_s - \boldsymbol{n}_{s-1} \right\|^2,$$

where $\boldsymbol{n}_i := \boldsymbol{y}_i^\ell / \|\boldsymbol{y}_i^\ell\|$ and we choose to normalize $\boldsymbol{y}^\ell$ since the difference $\|\boldsymbol{y}_s^\ell - \boldsymbol{y}_{s-1}^\ell\|^2$ would also penalize the norm of $\boldsymbol{y}_s^\ell$. A weighting coefficient $\zeta_C(s) > 0$ can change regularization strength along the sequence. The dependence of $\zeta_C$ on the sequence index can be derived from the covariance $\mathcal{K}_{s,t}$ (Appendix A.3) or chosen empirically. As expected, $\zeta_C$ is constant for uniform priors, and should monotonically increase for in-context learning problems.

In practice, adding this regularization term into the loss with some non-zero weight $w_C$ can incentivize the model to generate constant activations $\boldsymbol{y}_i^\ell$ with $\mathcal{R}_C = 0$, at least locally. A common way of stopping $\boldsymbol{y}$ from collapsing to a constant value is to adopt some form of contrastive learning approach. For example, inspired by the orthogonal projection loss (Ranasinghe et al., 2021), we regularize the scalar product of activations across samples in the batch for each sequence element independently:

$$\mathcal{R}_D \sim \sum_{s,\alpha,\beta} \zeta_D(s) \left( \boldsymbol{n}_s^{(\alpha)} \cdot \boldsymbol{n}_s^{(\beta)} - \delta_{\alpha,\beta} \right)^2,$$

where $\alpha$ and $\beta$ are the indices of two samples in the batch, $\delta_{\alpha,\beta}$ is the delta function and $\zeta_D(s) > 0$ can again be used to increase regularization strength towards the end of the sequence. In the following, we choose $\zeta_D = \zeta_C$. Notice that for $\alpha = \beta$, the dot product equals to $1 = \delta_{\alpha,\alpha}$ due to normalization, but for $\alpha \neq \beta$ the regularizer "pushes" the dot product of two different vectors $\boldsymbol{n}_s$ towards 0 making them orthogonal. We refer to regularizers that do not depend on cross-element correlations as *element-wise*. This particular regularizer is designed to favor orthogonality of sample representations within the batch and it proved to be sufficiently effective in our experiments, where we end up optimizing the joint loss

$$\mathcal{L}' = \eta \mathcal{L}_{\text{ce}} + (1 - \eta)\mathcal{L}_{\text{aux}} + w_C \mathcal{R}_C + w_D \mathcal{R}_D. \tag{3}$$

## 4 EXPERIMENTS

### 4.1 DATASETS

In this section, we describe 3 dataset families used in our experiments: (a) *synthetic* dataset with each sequence containing multiple arithmetic in-context learning tasks (numbers represented with digits), each of which could be resolved approximately by solving a system of two linear equations; (b) simple *linear regression* dataset with sequences containing sets of $(\boldsymbol{x}, \boldsymbol{y})$ pairs with $\boldsymbol{y}$ being a noisy linear function of $\boldsymbol{x}$; (c) *text mixture* dataset based on frequently used `wikipedia` (Wikimedia Foundation) and `c4` (Raffel et al., 2020) datasets, where we combine two random excerpts to form a single training example.

> 154*709=+07058|648*011=+05920|526*187=+06230|893*495=+11997#
> 122*395=-00273|827*301=+00526|216*082=+00134|399*879=-00480#
> 913*075=+01063|748*228=+01204|508*205=+00918|186*523=+01232#

Figure 2: An example of synthetic sequences analyzed in Sec. 4.2 with $n_{\text{tasks}} = 3$ tasks of $n_{\text{ex}} = 4$ examples each. Each example shows two $d$-digit integers ($A_{i,j}$ and $B_{i,j}$) and their truncated linear combination $C_{i,j} := \lfloor a_i A_{i,j} + b_i B_{i,j} \rfloor$, where $a_i \sim \mathcal{U}[0, 10)$, $b_i \sim \mathcal{U}(-10, 10)$ are the hidden task parameters, and $C_{i,j}$ is a $(d + 2)$-digit integer. "|" separates examples within tasks, "#" separates tasks. Line breaks are for visual clarity; the actual input is a single continuous sequence.

|  | Baseline | CGT (no Aux/Reg) | CGT (Aux) | CGT (Aux + Reg) |
|---|---|---|---|---|
| *Base Accuracy* | $77.7\% \pm 0.8\%$ | $\mathbf{79.0}\% \pm 0.8\%$ | $77.5\% \pm 1.3\%$ | $78.6\% \pm 1.2\%$ |
| *Specialized Accuracy* | – | $15.3\% \pm 7.2\%$ | $77.0\% \pm 1\%$ | $\mathbf{77.5}\% \pm 0.9\%$ |
| *Represen. Variation*, eq. (4) | – | $0.80$ | $0.39$ | $\mathbf{0.07}$ |
| *Linear Fit Error* | – | $0.19$ | $0.12$ | $\mathbf{0.04}$ |

Table 1: Performance of CGT with varying auxiliary loss and regularization on in-context learning tasks. Base accuracy uses dynamic context, while specialized accuracy freezes the context. See equation 4 for representation variation details. Values show mean and $3\sigma$ error.

**Synthetic In-Context Learning Setup.**   This synthetic dataset consists of sequences, each containing several ($n_{\text{tasks}} \geq 1$) in-context learning tasks. Each task is defined by two real-valued hidden parameters $(a, b)$ drawn uniformly from $[0, 10)$ and $(-10, 10)$. Specifically, each task $i$ consists of $n_{\text{ex}}$ examples. Each example $j$ within task $i$ presents two random $d$-digit integers, $A_{i,j}$ and $B_{i,j}$ (typically $d = 3$), and their truncated linear combination $C_{i,j} := \lfloor a_i A_{i,j} + b_i B_{i,j} \rfloor$, whcih is a signed $(d + 2)$-digit integer. The hidden coefficients $a_i$ and $b_i$ remain constant within a task. The model's goal is to infer $a_i$ and $b_i$ from the provided examples and then apply this linear function to new $d$-digit input numbers.

In most of our experiments, we used $n_{\text{tasks}} = 4$ and provided $n_{\text{ex}} = 4$ examples for each task. All examples were separated by a special token and all tasks within a sequence were separated by a different special token. A typical example with $a = 1$ and $b = 1$ could look like `012*023=+00035` and the same arguments for an example with $a = 0.5$, $b = -1.5$ would result in `012*023=-00028`. In the following, we will refer to substrings following "=" (`+00035` and `-00028` in the examples above) as *answers*. Examples of actual sequences are presented in Fig. 2.

Inferring $a$ and $b$ requires at least two examples. However, because the results are rounded, accurately determining $a$ and $b$ becomes more challenging. Thus, even powerful models likely need many examples to achieve near-perfect next-token prediction accuracy.

**Linear Regression.**   This dataset contained sequences with interleaved $(\boldsymbol{x}, \boldsymbol{y})$ pairs. Specifically, each sample $c_i$ contained a sequence of embeddings $(\boldsymbol{x}_1^i, \boldsymbol{y}_1^i, \ldots, \boldsymbol{x}_N^i, \boldsymbol{y}_N^i)$ with $\boldsymbol{y}_k^i := \boldsymbol{U}^i \boldsymbol{x}_k^i + \boldsymbol{b}^i + \epsilon \boldsymbol{q}_k^i \in \mathbb{R}^{d_{\text{out}}}$, $\boldsymbol{U}^i \in \mathbb{R}^{d_{\text{out}} \times d_{\text{in}}}$ being a random per-sample matrix, $\boldsymbol{x}_k^i \in \mathbb{R}^{d_{\text{in}}}$ and $\boldsymbol{b}^i, \boldsymbol{q}_k^i \in \mathbb{R}^{d_{\text{out}}}$ being random vectors. All components of these vectors and $\boldsymbol{U}^i$ were randomly sampled from a univariate Gaussian distribution $\mathcal{N}(0, 1)$. Before being passed to a Transformer, each $\boldsymbol{x}$ and $\boldsymbol{y}$ vector was zero-padded to the input embedding size and the positional encodings were added to them. The model was trained with an $L_2$ loss matching outputs at $\boldsymbol{x}_k^i$ positions with the corresponding $\boldsymbol{y}_k^i$ values. In most of our experiments, we used $d_{\text{in}} = 16$, $d_{\text{out}} = 1$ and $\epsilon = 0.1$.

**Text Mixture Datasets.**   In another set of experiments, we use text datasets such as `wikipedia` and `c4`. Our models are trained on sequences constructed from individual text samples or combinations of two independent text samples coming from the source dataset. When two input text samples are concatenated to form a single sequence, we cut the first text excerpt at a random position sampled uniformly from the range $[l_{\text{start}}, l_{\text{finish}}]$ and concatenate the second text sequence to it. The concatenation is done after both text sequences are tokenized and the final produced sample is truncated at the maximum sequence length $l_{\text{max}}$. In most of our experiments, the total sequence length was $l_{\text{max}} = 512$, $l_{\text{start}} = 256$ and $l_{\text{finish}} = 384$. For additional information see Appendix B.4.

### 4.2 In-Context Learning Results

Our first experiments were conducted with the synthetic in-context learning dataset described in Section 4.1 with $n_{\text{tasks}} = 4$ tasks, $n_{\text{ex}} = 4$ examples per task and $d = 3$. We based our CGT models on the GPT2 Transformer architecture (Radford et al., 2019) and trained them with the loss given by equation 3, a combination of cross-entropy loss, our auxiliary loss and the element-wise regularization (Appendix B.1 for model details). Baseline models had 6 layers, an inner dimension of $\dim \boldsymbol{x} = 112$, and 7 heads. The CGT model also introduced $\boldsymbol{y}$ with $\dim \boldsymbol{y} = 64$ allocating 4 additional heads to it. After a brief hyper-parameter tuning stage, we chose $\ell = 4$, $w_C = 0.08$ and $w_D = w_C/2$. Appendix B.2 summarizes additional parameters and ablation studies.

**Model performance.**   First we compared the performance of four different models: (a) a baseline $\boldsymbol{x}$-only model that corresponds to a traditional Transformer, (b) CGT model with both $\boldsymbol{x}$ and $\boldsymbol{y}$ (cross-entropy loss only, $\eta = 1$), (c) CGT with auxiliary loss ($\eta = 0.5$) and (d) CGT with $\eta =$

0.5, auxiliary loss and element-wise regularization. In our experiments, we found that regularizing the sequence uniformly with $\zeta_C(s) = \text{const}$ results in the same performance as if we prioritized regularization in the last two examples. All models were trained from scratch with 6 independent runs per experiment. We compared model accuracies on the answers[4] in the last two examples (of 4) for each individual task in all test sequences. When evaluating CGT models, we also compared: (a) inference with dynamic per-token $\boldsymbol{y}^\ell$ and (b) inference with specialized models obtained by freezing $\boldsymbol{y}^\ell$ at the end of first two examples and removing the first two examples from context. Our experiments with *task vectors* in baseline models with activation transplantation on "|" and "=" tokens reached the top accuracies of $36.8\%$ and $40.7\%$ correspondingly, suggesting that task vectors do not typically emerge in our baseline experiments.

Results presented in Table 1 can be interpreted as follows. Extending a baseline model to CGT by introducing additional $\boldsymbol{y}$ activations and $\boldsymbol{y}^\ell$-dependent transformations $\mathbf{T}^\kappa(\boldsymbol{y}^\ell)$ increases model accuracy from $77.7\%$ to $79.0\%$ when training with cross-entropy loss alone. However, specializing these CGT models (by freezing $\boldsymbol{y}^\ell$ and removing first two examples from the context) leads to severe performance degradation. This suggests that the dynamics of $\boldsymbol{y}^\ell$ and the information encoded in its token-to-token change, is crucial for proper operation of these trained models.

Once we add auxiliary loss ($\eta = 0.5$) as discussed in Section 3.2, the average accuracy of specialized models reaches $77.0\%$ approaching the performance of the baseline model. Finally, when adding element-wise regularization, we observe the specialized model accuracy to increase further to $77.5\%$ (while also improving the average model performance with dynamic $\boldsymbol{y}^\ell$). We thus conclude that (a) our generated task-specialized models successfully solve the task without any examples in context reaching nearly the same accuracy as the models seeing previous demonstrations in-context, (b) element-wise regularization enforcing the smoothness and sequence-to-sequence variability of $\boldsymbol{y}^\ell$ has a positive impact on model performance (with and without $\boldsymbol{y}^\ell$ freezing).

$\boldsymbol{y}^\ell$ **as task representation.**  We also analyzed the properties of the context representation $\boldsymbol{y}^\ell$. First we computed the variation of $\boldsymbol{y}^\ell$ on the last two examples of 4 (at which point the model should have inferred the task at hand). Our variation metric was chosen as:

$$\frac{1}{|E_2|} \sum_{s \in E_2} \|\bar{\boldsymbol{y}}_s - \bar{\boldsymbol{y}}_{s-1}\|, \tag{4}$$

where $E_1$ and $E_2$ denote the sequence segments of the first and the last two examples correspondingly and $\bar{\boldsymbol{y}}^\ell := \delta\boldsymbol{y}^\ell / \sqrt{\langle\|\delta\boldsymbol{y}^\ell\|^2\rangle_{E_1+E_2}}$ with $\delta\boldsymbol{y}^\ell := \boldsymbol{y}^\ell - \langle\boldsymbol{y}^\ell\rangle_{E_1+E_2}$. In other words, we compute the total variation of $\bar{\boldsymbol{y}}^\ell(\boldsymbol{t})$ on $E_2$ with $\bar{\boldsymbol{y}}^\ell$ being normalized across all 4 examples (on $E_1 + E_2$). Results presented in Table 1 confirm that element-wise regularization leads to a significant reduction in $\boldsymbol{y}^\ell$ variation.

We then verified that $\boldsymbol{y}^\ell$ does in fact encode information about the task by training a *linear model* that predicted normalized values of task multipliers $a$ and $b$ from the context embedding $\boldsymbol{y}_s^\ell$ computed at arbitrary $s \in E_2$. A typical linear fit for both of these coefficients in a model with element-wise regularization is illustrated in Figure 3(b). Quite surprisingly, both coefficients can be approximated by a linear function of $\boldsymbol{y}^\ell$ to a very high accuracy. We compared the mean error of this linear fit (on the last two examples) across 4 models (see Table 1), confirming again that the model with element-wise regularization was characterized by the best linear fit.

The effect of element-wise regularization can be studied further by training CGT model with regularization, but no auxiliary loss ($\eta = 1$). Figure 3(a) shows a typical plot of the average dot-product $\langle \boldsymbol{n}_s \cdot \boldsymbol{n}_t \rangle$ of normalized context embeddings $\boldsymbol{n}_s = \boldsymbol{y}_s^\ell / \|\boldsymbol{y}_s^\ell\|$ emerging in CGT models. A block-diagonal structure with 4 blocks (of nearly orthogonal $\boldsymbol{n}$ values) reflects the fact that there are 4 independent tasks in each sequence. Notice that the slow embedding generally evolves in the first half of each task (first 30 tokens), when the model processes the first two examples, but then stabilizes and hardly changes on the last two examples.

### 4.3 LINEAR REGRESSION RESULTS

Just like with the previous synthetic in-context learning dataset, we observed that CGT models trained with equation 3 and specialized by freezing $\boldsymbol{y}^\ell$ after 5, 10 and 20 examples achieved linear

---

[4]Digits following "=" in each example.

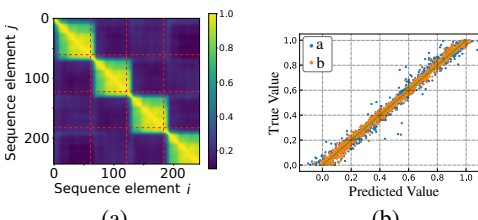
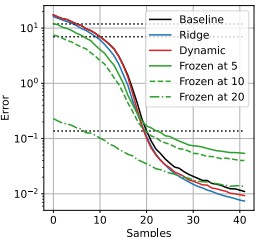

(a)       (b)

Figure 3: **(a)** The dot product $\boldsymbol{n}_i \cdot \boldsymbol{n}_j$ plot for normalized $\boldsymbol{y}^\ell$ embeddings at two different locations in the sequence with 4 tasks and 4 examples per task; **(b)** linear regression results for the multipliers $a$ and $b$ given the average value of $\boldsymbol{y}^\ell$, the plot shows agreement between predicted and groundtruth values.

Figure 4: The model continuously learns and improves its predictions as it receives more samples. Models: (i) "baseline" Transformer ($\dim \boldsymbol{x} = 128$); (ii) CGT (dynamic $\boldsymbol{y}^\ell$, $\dim \boldsymbol{x} = 128$, $\dim \boldsymbol{y} = 64$); (iii) CGT (frozen $\boldsymbol{y}^\ell$ after 5, 10, or 20 samples). Horizontal lines: Baseline $L_2$ errors after 5, 10 and 20 samples. Ridge regression error is also shown.

regression accuracy comparable to in-context presentation of the same examples. While 5 and 10-example specialization yielded near-optimal accuracy largely independent of model parameters, 20-example specialization proved to be more sensitive. Three parameters were particularly important: the input dimension ($\dim \boldsymbol{x}$), the output dimension ($\dim \boldsymbol{y}$), and the layer index $\ell$ at which $\boldsymbol{y}^\ell$ is applied. Increasing $\dim \boldsymbol{x}$ improved overall performance as the number of in-context examples increased. As expected, larger $\dim \boldsymbol{y}$ was important for specialized model accuracy. Perhaps more surprisingly, smaller values of $\ell$ significantly degraded model performance. Only when choosing $\ell$ to be just one layer below the final layer (with $\boldsymbol{y}^\ell$ modulating a single layer), were we able to see a specialized model approach the optimal accuracy (see Fig. 4). Using monotonically growing $\zeta_C$ profiles (including quadratic motivated by equation 1) did not have a statistically significant impact on model performance. Details of our experiments are provided in Appendix D.2.

## 4.4 Text Mixture Results

In most of our experiments with text datasets, each sequence was a mixture of two distinct c4 text excerpts. The CGT model was based on a 12-layer version of GPT-2 with $\ell = 8$ being the layer for reading out $\boldsymbol{y}^\ell$. We used the loss given by equation 3 with $w_C = 0.04$ and $w_D = w_C/2$. Additional model parameters are discussed in Appendices B.2 and B.4.

**Model performance.** First we verified that the addition of a context embedding $\boldsymbol{y}$ in CGT architecture improves performance on c4 text dataset (Fig. 5). Using main embeddings with dimensionality $\dim \boldsymbol{x} = 128$, adding a 128-dimensional context embedding ($\dim \boldsymbol{y} = 128$) at layer $\ell = 8$ yielded a substantial improvement of 0.23 in average cross-entropy loss from a baseline of approximately 3.65. For comparison, using main embeddings with $\dim \boldsymbol{x} = 224$ and a smaller, 32-dimensional context embedding ($\dim \boldsymbol{y} = 32$) at the same layer resulted in a smaller improvement of 0.04 from a baseline of 3.25.

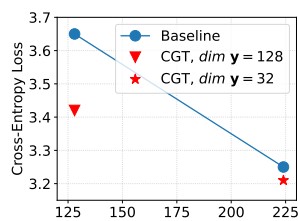

Figure 5: Cross-entropy loss on c4 text dataset.

While the improvement achieved with $\dim \boldsymbol{x} = 128$ and $\dim \boldsymbol{y} = 128$ is less than that observed when simply increasing the main embedding dimensionality proportionally to 224 (roughly equivalent to $\dim \boldsymbol{x} + \dim \boldsymbol{y}$ in a conventional model), the computational overhead of introducing $\boldsymbol{y}^\ell$ at a later layer ($\ell = 8$) is relatively small.

Next, we evaluated the performance of specialized models generated from c4 pre-trained models by freezing the context representation $\boldsymbol{y}^\ell$ after processing an initial portion of the text. We split 11 600 validation samples into two parts, typically at the end of the first sentence ( around 400 characters or 100 tokens). After processing the first part, $\boldsymbol{y}^\ell$ was frozen and the specialized CGT processed the second part. We calculated the average per-token cross-entropy loss across all samples.

Initially, the specialized model had lower loss, but the non-specialized model caught up and surpassed it after roughly 200 tokens (see Fig. 6(a)). This suggests that the fixed $\boldsymbol{y}^\ell$ benefited the specialized model near its point of calculation but hindered performance further down the sequence,

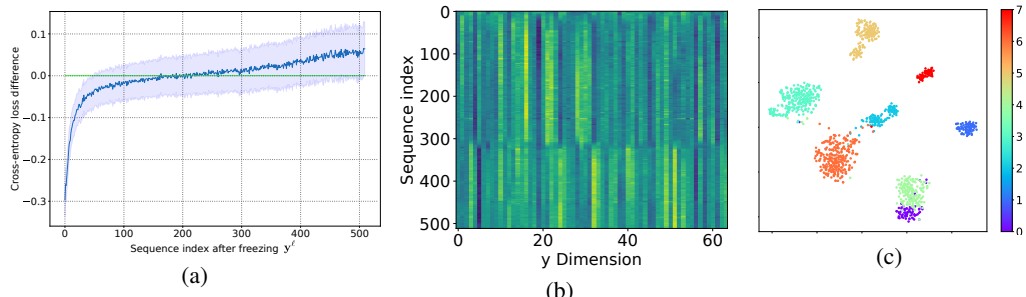

Figure 6: **(a)** Average difference between the cross-entropy losses of the specialized (fixed $\boldsymbol{y}^\ell$, blue curve) and non-specialized (dynamic $\boldsymbol{y}^\ell$, green curve) models (typical loss value is $\sim 3$); the specialized model is better where this difference is below zero. **(b)** Evolution of $\boldsymbol{y}^\ell$ along the sequence containing two `c4` text excerpts joined at $305$. **(c)** $t$-SNE plot for `wikipedia` articles from 8 different categories.

likely due to thematic shifts and the fact that we trained the model on small subsequences containing two separate text excerpts. Similar behavior was observed when comparing the specialized model to a separately trained baseline model, with performance leveling around 300 tokens (see Fig. 13(a) in Appendix).

To address this, we experimented with updating $\boldsymbol{y}^\ell$ as a moving average of its values computed on the second part of the text, rather than simply clamping it. This yielded a consistently lower cross-entropy loss across the entire sequence (Figure 14 in Appendix). This approach effectively injects information from the first part of the text into the second while allowing $\boldsymbol{y}^\ell$ to adapt to the changing context. Thus, $\boldsymbol{y}^\ell$ can be viewed as a memory state, or *topic vector* that we can flexibly manipulate.

$\boldsymbol{y}^\ell$ **representation.** Our model, trained with element-wise regularization, learns to encode textual topics in the latent variable $\boldsymbol{y}^\ell$. This is evident in two ways. First, $\boldsymbol{y}^\ell$ exhibits clear transitions between different text excerpts within a single sample (see Fig. 6(b)).

Second, $\boldsymbol{y}^\ell$ serves as a meaningful embedding for documents. We calculated $\boldsymbol{y}^\ell$ across hundreds of `wikipedia` pages from 8 distinct categories (see details in Appendix B.4). The $t$-SNE (van der Maaten & Hinton, 2008) plot in Fig. 4.4 shows the clustering of pages from the same categories, with noticeable distinction between different categories, except for "Mathematical identities" and "Theoretical physics," which aligns with their semantic similarity. This behavior of $\boldsymbol{y}^\ell$ is critically dependent on using element-wise regularization and does not generally emerge without it (see Fig. 16). Moreover, we assessed our model on various out-of-distribution mixtures of 3 text excerpts, observing transitions of $\boldsymbol{y}^\ell$ within approximately 10 to 20 tokens from the merging locations (see Fig. 13(b) in Appendix).

In our experiments with VAE model discussed in Appendix D.4, we observed the emergence of embeddings with similar properties. As expected, by controlling $\beta_y$ (see equation 2) and the autocorrelation length ($\lambda$ in Appendix B.2.3), we were able to vary the smoothness of the emergent context representation $\boldsymbol{y}_s^\ell$.

## 5 CONCLUSIONS

Learning slow features that carry information about the global context in a sequence is important for understanding and interpreting data. Here we propose an approach for incentivizing a Transformer model to discover such slow representation within its inner activations. We then modify the model architecture to parameterize local computation by these learned slow features, showing that it is then possible to generate models that are uniquely specialized to a particular local context and no longer need to have direct access to it. While we only consider several simple examples in our experiments (a synthetic few-shot in-context learning task, linear regression and a mixture of texts), we believe that this approach can prove useful for representation learning, model interpretability and generation of specialized models.

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

# Appendix

## A  ADDITIONAL APPROACHES

### A.1  VAE APPROACH

One approach to incorporating the *slowness prior* into the model is to view our Transformer model as a Variational Autoencoder (Kingma & Welling, 2014) with a Gaussian process prior on $\boldsymbol{y}^\ell$.

In this setup, we assume that the probability distribution over the token sequence $\boldsymbol{t}$ can be represented as $\int p_\phi(\boldsymbol{t}|\boldsymbol{z}^\ell)p_0(\boldsymbol{z}^\ell)\,d\boldsymbol{z}^\ell$ with $p_\phi(\boldsymbol{t}|\boldsymbol{z}^\ell)$ being a *causal decoder*, parameterized by $\phi$, and $p_0(\boldsymbol{z}^\ell) = p_0(\boldsymbol{x}^\ell)p_0(\boldsymbol{y}^\ell)$ being the prior. Following a conventional Variational Autoencoder setup (Kingma & Welling, 2014), we can approximate the true distribution $p_\phi(\boldsymbol{z}^\ell|\boldsymbol{t})$ with a variational distribution $q_\psi(\boldsymbol{z}^\ell|\boldsymbol{t})$, parameterized by $\psi$. Using the evidence lower bound (ELBO), we can then derive an objective function:

$$\mathcal{L} = \mathbb{E}_{\boldsymbol{t}\sim p(\boldsymbol{t})}\Big[\mathbb{E}_{\boldsymbol{z}^\ell\sim q_\psi(\boldsymbol{z}^\ell|\boldsymbol{t})}\log p_\phi(\boldsymbol{t}|\boldsymbol{z}^\ell) + D_{\mathrm{KL}}(q_\psi(\boldsymbol{z}^\ell|\boldsymbol{t})|p_0(\boldsymbol{z}^\ell))\Big].$$

In the following, we assume statistical independence of $\boldsymbol{x}^\ell$ and $\boldsymbol{y}^\ell$ in $q_\psi(\boldsymbol{z}^\ell|\boldsymbol{t})$ and adopt the $\beta$-VAE approach relying on two independent constraints, on $\boldsymbol{x}^\ell$ and $\boldsymbol{y}^\ell$ resulting in:

$$\mathcal{L} = \mathbb{E}_{\boldsymbol{t}\sim p(\boldsymbol{t})}\Big[\mathbb{E}_{\boldsymbol{z}^\ell\sim q_\psi(\boldsymbol{z}^\ell|\boldsymbol{t})}\log p_\phi(\boldsymbol{t}|\boldsymbol{z}^\ell) + \beta_x D_{\mathrm{KL}}(q_\psi(\boldsymbol{x}^\ell|\boldsymbol{t})|p_0(\boldsymbol{x}^\ell)) +$$
$$+ \beta_y D_{\mathrm{KL}}(q_\psi(\boldsymbol{y}^\ell|\boldsymbol{t})|p_0(\boldsymbol{y}^\ell))\Big]. \quad (5)$$

While it could be useful to define a prior on $\boldsymbol{x}^\ell$, in the following we choose $\beta_x = 0$ and let $\boldsymbol{x}^\ell$ be unconstrained, only constraining our slow activations $\boldsymbol{y}^\ell$. As a result, we can view the first term in equation 5 as a conventional autoregressive sequence reconstruction loss, while the last KL divergence term acts as a regularizer on $\boldsymbol{y}^\ell$.

We can simplify our analysis further by choosing a naive[5] *causal encoder* $q_\psi(\boldsymbol{z}^\ell|\boldsymbol{t})$:

$$q_\psi(\boldsymbol{z}^\ell|\boldsymbol{t}) \propto \prod_{i=1}^{d}\exp\left[-\sum_{s=1}^{n}\frac{(y_{i,s}^\ell - \mu_{i,s}^y(\boldsymbol{t}))^2}{2\sigma_{i,s}(\boldsymbol{t})^2}\right]\delta(x_{i,s}^\ell - \mu_{i,s}^x(\boldsymbol{t})), \quad (6)$$

effectively treating elements $\boldsymbol{y}_s^\ell$ taken at different positions $s$ as statistically independent draws from corresponding Gaussian distributions. As a result, our final $\beta$-VAE loss takes the following form:

$$\mathcal{L}_{\mathrm{VAE}}(\boldsymbol{t}) = \mathcal{L}_{\mathrm{rec}} - \frac{\beta_y}{2}\sum_{i,s}\log\sigma_{i,s}(\boldsymbol{t}) + \frac{\beta_y}{2}\sum_{i,s,t}\mathcal{K}_{s,t}^{-1}\mu_{i,s}(\boldsymbol{t})\mu_{i,t}(\boldsymbol{t}) + \frac{\beta_y}{2}\sum_{i,s}\mathcal{K}_{s,s}^{-1}\sigma_{i,s}(\boldsymbol{t}), \quad (7)$$

where $\beta_y > 0$ is the $D_{\mathrm{KL}}$ term weighting coefficient and $\mathcal{L}_{\mathrm{rec}} = \mathcal{L}_{\mathrm{ce}}$ is the reconstruction cross-entropy loss. In our experiments, we use the modified loss $\eta\mathcal{L}_{\mathrm{ce}} + (1-\eta)\mathcal{L}_{\mathrm{aux}}$ in place of $\mathcal{L}_{\mathrm{rec}}$ to include our auxiliary loss. Notice that $\mathcal{K}^{-1}$ can be precomputed making this calculation sufficiently low-cost.

This formulation allows us to view the full Transformer model as a combination of two parts: an encoder $q_\psi(\boldsymbol{z}^\ell|\boldsymbol{t})$ mapping the input $\boldsymbol{t}$ to intermediate activations $\boldsymbol{z}^\ell$ at some layer $\ell$, and a *decoder* $p_\phi(\boldsymbol{t}|\boldsymbol{z}^\ell)$ reconstructing the input from these latent variables. Choosing causal Transformer layers for parameterizing both $p_\phi$ and $q_\psi$, our model differs from a standard Transformer only in that its activations $\boldsymbol{z}^\ell$ are no longer deterministic.

The KL divergence term[6] in equation 7 can be seen to penalize very large and very small values of $\sigma_s$ and non-zero $\mu_s$. The regularization effect on $\mu$ can be studied by computing eigenvectors of $\mathcal{K}^{-1}$. For example, consider $\mathcal{K}_{s,t} \sim \exp(-|s-t|/\lambda)$. For sufficiently large $\lambda$, the eigenvalues can typically be seen to grow rapidly with the number of oscillations in the corresponding eigenvectors, highlighting the fact that this regularization term suppresses rapid fluctuations uniformly along

---

[5]recall that $q_\psi(\boldsymbol{z}^\ell|\boldsymbol{t})$ being unable to represent the true $p_\phi(\boldsymbol{z}^\ell|\boldsymbol{t})$ hurts the bound

[6]all except the first term in the right-hand side of equation 7

the sequence. Conversely, for $\mathcal{K}_{s,t}$ more characteristic for few-shot in-context learning tasks (see equation 1), the strongest regularized eigenvectors are localized at the end of the sequence. In other words, this $D_{\mathrm{KL}}$ term regularizes significant changes at the end of the $\boldsymbol{y}^\ell$ sequence more severely, respecting the prior that expects most changes to be localized to first few examples.

## A.2 Distribution-Matching Regularizers

The VAE approach outlined above allows us to incorporate a Gaussian process prior on $\boldsymbol{y}^\ell$ in a natural way (in the following we drop index $\ell$ for brevity). Here we outline a different method enforcing a similar prior, namely that $(\boldsymbol{y}_1, \ldots, \boldsymbol{y}_n)$ adhere to a chosen $p(\boldsymbol{y}_1, \ldots, \boldsymbol{y}_n)$. Since estimating probability distribution of a high-dimensional random process is typically complicated, we need to rely on a simpler approach. Specifically, we consider a sufficiently flexible parametric family $p_\theta$ and then regularize the values of the parameter estimators $\hat{\theta}(\boldsymbol{y})$ to be equal to their predefined values by using, for example, a regularizer

$$\mathcal{R}_P \sim \left\| \hat{\theta}(\boldsymbol{y}) - \theta_0 \right\|^2. \tag{8}$$

Here we utilize a naïve $L_2$ regularization of the distribution parameters, but other choices could also be considered.

**Gaussian process example.** Instead of regularizing the derivative of $\boldsymbol{y}_s$, here we introduce a more natural constraint on $\boldsymbol{y}$ requesting that these slow activations are a stationary Gaussian process with zero mean and kernel $\mathcal{K}$ depending only on the relative position of two elements in the sequence. Different choices of $\mathcal{K}$ can control how slowly $\boldsymbol{y}_s$ is expected to change along the sequence.

Assuming that $\boldsymbol{y}$ is a multi-variate Gaussian distribution, we can estimate the mean and covariance matrix:

$$\boldsymbol{\mu}_s = \langle \boldsymbol{y}_s \rangle \quad \text{and} \quad \Sigma_{s,t} = \langle (\boldsymbol{y}_s - \boldsymbol{\mu}_s)(\boldsymbol{y}_t - \boldsymbol{\mu}_t) \rangle,$$

where the averaging is performed over the batch of samples. Remembering our Gaussian process assumption, we can then expect that $\boldsymbol{\mu}_s = 0$ and $\Sigma_{s,t,i,j} = \mathcal{K}(|s - t|)\delta_{i,j}$, which we can enforce by utilizing the regularizer equation 8:

$$\mathcal{R}_P \sim \frac{1}{N} \sum_s \|\boldsymbol{\mu}_s\|^2 + \frac{1}{N^2} \sum_{s,t,i,j} \left( \langle \Delta y_{s,i} \Delta y_{t,j} \rangle_\alpha - \mathcal{K}_{|s-t|} \delta_{i,j} \right)^2,$$

where $N$ is the total number of elements in each sequence and $\Delta \boldsymbol{y}_s := \boldsymbol{y}_s - \boldsymbol{\mu}_s$. Here $\langle \cdot \rangle$ denotes averaging over individual samples in the batch. Notice that in practice, we can reduce the cost of the proposed computation by sampling only a small set of all possible sequence elements $(s, t)$ or embedding dimensions $(i, j)$.

Notice that we can also use a simplified form of this regularizer, where we remove constraints on cross-token correlations:

$$\mathcal{R}'_D \sim \sum_s \left[ \|\langle \boldsymbol{y}_s \rangle\|^2 + \sum_{i,j} \left( \langle \Delta y_{s,i} \Delta y_{s,j} \rangle - \delta_{i,j} \right)^2 \right],$$

where $\Delta \boldsymbol{y} := \boldsymbol{y} - \langle \boldsymbol{y} \rangle$ and $\langle \cdot \rangle$ denotes averaging over individual elements in a batch. Compared to the orthogonal projection loss used in Section 3.3, here we instead compute and regularize sample statistics.

## A.3 Towards Element-Wise Regularizers

The VAE loss 7 is regularizing mean $\boldsymbol{\mu}_s$ via a term proportional to:

$$\sum_{i,s,t} \mathcal{K}_{s,t}^{-1} \mu_{i,s}(\boldsymbol{t}) \mu_{i,t}(\boldsymbol{t}). \tag{9}$$

This regularizer minimized only when $\boldsymbol{\mu}_s = 0$ is counteracting the need to propagate information via the latent variable $y_{i,s}^\ell \sim \mathcal{N}(\mu_{i,s}, \sigma_{i,s})$, so that the input sequence can be properly reconstructed by the decoder ($\boldsymbol{\sigma}$ cannot go to zero due to other regularization terms).

But on top of regularizing $\|\boldsymbol{\mu}_s\|$, equation 9 can also be seen to penalize rapid $\boldsymbol{\mu}_s$ changes. One way of seeing this is to consider a continuous limit of equation 9 for a single-component $\mu$:

$$\Gamma := \int_0^n ds \int_0^n dt \, \mathcal{K}^{-1}(s,t)\mu(s)\mu(t).$$

Introducing $\tau := (s+t)/2$ and $\delta := s-t$, we can rewrite this integral as:

$$\int_V \mathcal{K}^{-1}(\tau+\delta/2, \tau-\delta/2)\mu(\tau+\delta/2)\mu(\tau-\delta/2)\, d\tau\, d\delta$$

with the integration volume $V$ being given by a "rhombus" $\tau \in [0,n]$ and $\delta(\tau) \in [\max(-2\tau, -2(n-\tau)), \min(2\tau, 2(n-\tau))]$. In the following, we will ignore the volume boundaries and integrate over the whole $\mathbb{R}^2$.

If $\Lambda(\tau, \delta) := \mathcal{K}^{-1}(\tau+\delta/2, \tau-\delta/2)$ quickly decays with increasing $|\delta|$ as we step away from the diagonal of $\mathcal{K}^{-1}$, we can approximate:

$$\Gamma \approx \int_V \Lambda(\tau, \delta)\left(\mu(\tau) + \mu'(\tau)\frac{\delta}{2} + O(\delta^2)\right)\left(\mu(\tau) - \mu'(\tau)\frac{\delta}{2} + O(\delta^2)\right)\, d\tau\, d\delta,$$

or simply

$$\Gamma \approx \int \kappa_0(\tau)\mu^2(\tau)\, d\tau - \frac{1}{4}\int \kappa_2(\tau)\mu'^2(\tau)\, d\tau, \tag{10}$$

where $\kappa_m(\tau) := \int_0^\infty \Lambda(\tau, \delta)\delta^m\, d\delta$. The second term in this approximation can be seen to regularize the derivative of $\mu$ since $\kappa_2(\tau)$ is generally negative.

For uniform kernels including $\mathcal{K}_{s,t} \sim \exp(-|s-t|/\lambda)$, the corresponding $\Lambda(\tau, \delta)$ is independent of $\tau$ almost everywhere (except close to $\tau=0$ and $\tau=n$ in a finite region $V$), and the regularizer $\Gamma$ can be simply replaced with an $L_2$ regularization of the first derivative of $\mu$. For non-uniform kernels $\mathcal{K}$, the coefficient $\kappa_2(\tau)$ needs to be pre-computed analytically or empirically.

Seeing the role of the regularizer $\Gamma$, we can try replacing a complex VAE regularization scheme with a much simpler "element-wise" regularizer in a conventional Transformer model by choosing it to be proportional to:

$$-\sum_{s=2}^n \kappa_2(s)\left\|\boldsymbol{y}_s^\ell - \boldsymbol{y}_{s-1}^\ell\right\|^2.$$

In the absence of noise injection characteristic for VAEs, penalizing the norm of $\|\boldsymbol{y}_s^\ell\|$ can be detrimental to model performance and hence we choose to regularize the derivative of the normalized representation $\boldsymbol{n}_i := \boldsymbol{y}_i^\ell/\|\boldsymbol{y}_i^\ell\|$:

$$\mathcal{R}_C^\ell = \sum_{s=2}^n \zeta_C(s)\left\|\boldsymbol{n}_s - \boldsymbol{n}_{s-1}\right\|^2$$

with $\zeta_C(s) \sim -\kappa_2(s)$ in order to match regularization in equation 10.

## B  MODEL DETAILS AND PARAMETERS

### B.1  MODEL DETAILS

In all of our experiments, we used GPT-2 style Transformer models with GELU nonlinearities.

Each MLP layer separated $\boldsymbol{x}$ and $\boldsymbol{y}$ transformations, effectively using two MLPs for processing $\boldsymbol{x}$ and $\boldsymbol{y}$ correspondingly (ignoring biases for brevity):

$$\boldsymbol{x}^{\nu+1} = \mathbf{W}_2^{\boldsymbol{x}}\sigma(\mathbf{W}_1^{\boldsymbol{x}}\boldsymbol{x}^\nu),$$
$$\boldsymbol{y}^{\nu+1} = \mathbf{W}_2^{\boldsymbol{y}}\sigma(\mathbf{W}_1^{\boldsymbol{y}}[\boldsymbol{x}^\nu, \boldsymbol{y}^\nu]),$$

where $[\cdot, \cdot]$ denotes vector concatenation, $\mathbf{W}_*^*$ are linear operators with corresponding matrices $\boldsymbol{W}_1^{\boldsymbol{x}} \in \mathbb{R}^{i_x \times d_x}$, $\boldsymbol{W}_2^{\boldsymbol{x}} \in \mathbb{R}^{d_x \times i_x}$, $\boldsymbol{W}_1^{\boldsymbol{y}} \in \mathbb{R}^{i_y \times (d_x+d_y)}$, $\boldsymbol{W}_2^{\boldsymbol{y}} \in \mathbb{R}^{d_y \times i_y}$. The inner dimensions were typically chose to be $i_x := 4d_x = 4\dim\boldsymbol{x}$ and $i_y := 4d_y = 4\dim\boldsymbol{y}$.

Similarly, each self-attention layer had separate $H_x$ heads acting on $\boldsymbol{x}$ alone and producing the final output that was completely $\boldsymbol{y}$-independent. Total of $H_y$ $(d_y/H_y)$-dimensional heads were reserved for self-attention on $\boldsymbol{y}$ with key/query/value vectors generated from the complete state $(\boldsymbol{x}, \boldsymbol{y})$ thus allowing $\boldsymbol{y}$ to absorb information from $\boldsymbol{x}$:

$$\boldsymbol{k^x} = \mathbf{K^x}\boldsymbol{x}, \qquad \boldsymbol{q^x} = \mathbf{Q^x}\boldsymbol{x}, \qquad \boldsymbol{v^x} = \mathbf{V^x}\boldsymbol{x},$$
$$\boldsymbol{k^y} = \mathbf{K^y}[\boldsymbol{x}, \boldsymbol{y}], \qquad \boldsymbol{q^y} = \mathbf{Q^y}[\boldsymbol{x}, \boldsymbol{y}], \qquad \boldsymbol{v^y} = \mathbf{V^y}[\boldsymbol{x}, \boldsymbol{y}],$$

where we omitted the computation stage index $\nu$ and the head index $h$ for brevity.

Each Transformer block contained self-attention layer followed by the MLP layer, as described above, with inner normalizaton operations applied separately to $\boldsymbol{x}$ and $\boldsymbol{y}$.

Before layer $\ell$, the computation on $\boldsymbol{x}$ was completely independent of $\boldsymbol{y}$, but at and after layer $\ell$, we applied an additional transformation $\mathbf{T}^\kappa(\boldsymbol{x}^\kappa; \boldsymbol{y}^\ell)$ on $\boldsymbol{x}^\kappa$ before each self-attention and each MLP operation.

### B.2    MODEL PARAMETERS

We trained our models using ADAM optimizer with the learning rate typically set to $2.5 \cdot 10^{-4}$ or $5 \cdot 10^{-4}$ for the total of $400{,}000$ steps with cosine learning rate decay (warmup of $10{,}000$ steps) and batch size of $128$. We used Google TPU v5e 4x4 as our training hardware platform, which took us to spend about 10 hours training the model. We did not use dropout in most of our experiments, which allowed us to reach higher accuracies in the in-context learning setup, but resulted in a degraded model stability: the final model accuracy for different initial seeds could differ by as much as $2\%$. High values of weight decay were also observed to hurt the model performance and we set it to $10^{-8}$ in most of our experiments.

#### B.2.1    IN-CONTEXT LEARNING.

In most of our experiments with the synthetic dataset, we used a 6-layer model with 7 to 11 self-attention heads. The baseline model had $h_x = 7$ heads with the embedding size of $d_x = 112$. The model with $d_y$-dimensional $\boldsymbol{y}^\ell$ used $7 + h_y$ heads, where $h_y = d_y/16$, making the total embedding size equal to $d_x + d_y = 112 + 16h_y$. In our experiments with specialized models, we chose $d_y = 64$ (and hence $h_y = 4$) and the rank of $\delta \boldsymbol{W}^\nu$ was 4 and $M = 16$ (total number of $\boldsymbol{L}$ and $\boldsymbol{R}$ matrices). We chose $w_C = 0.08$ and $w_D = 0.04$. The auxiliary loss was typically computed using the entire rest of the sequence or a small context of size $\Delta = 10$ (each answer contained only 7 tokens).

**Ablation studies.**    We conducted additional ablation studies varying four parameters:

1. layer $\ell$ where $\boldsymbol{y}^\ell$ is computed (Fig. 7(a)),
2. rank $r$ of the generated matrix (Fig. 7(b)),
3. values of $w_C$ and $w_D$ (Fig. 8),

All experiments measured the performance of specialized models with $\dim \boldsymbol{y}^\ell = 64$ with examples used to generate $\boldsymbol{y}^\ell$ presented *in context* (*first setup* in Sec. 4.2). While it is clear that confident statements require significantly more experiments for statistically significant results, we may draw some preliminary conclusions. First, the optimal location of layer $\ell$ appears to be at $\ell = 4$, not too close to the beginning where the model may not have enough time to produce an accurate context representation $\boldsymbol{y}^\ell$ and not too late, where there is little time to modulate the computation using $\boldsymbol{y}^\ell$. Secondly, models with generated rank-2 and rank-4 matrices appear to outperform models with rank-1 matrices. Increasing derivative regularization strength $w_C$ appears to hurt performance above $w_C = 0.04$. And increasing the orthogonal projection loss weight $w_D$ appears to not hurt model performance and possibly even improves it.

#### B.2.2    TEXT MIXTURE.

In our text experiments, we chose $w_C = 2w_D = 0.08$ and our models contained 12 layers with $\ell = 8$. The total number of tokens was equal to 8000 byte pair encoding subwords (Sennrich et al., 2015) and the total sequence size was $512$.

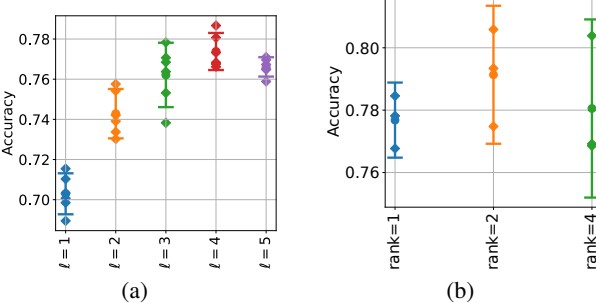

(a)                                    (b)

Figure 7: Ablation study results: **(a)** dependence of the model accuracy with frozen $\boldsymbol{y}^{\ell}$ on the layer index $\ell$ (model trained with the auxiliary loss and the element-wise regularization); **(b)** varying the rank of the generated matrices with $w_C = 2w_D = 0.08$.

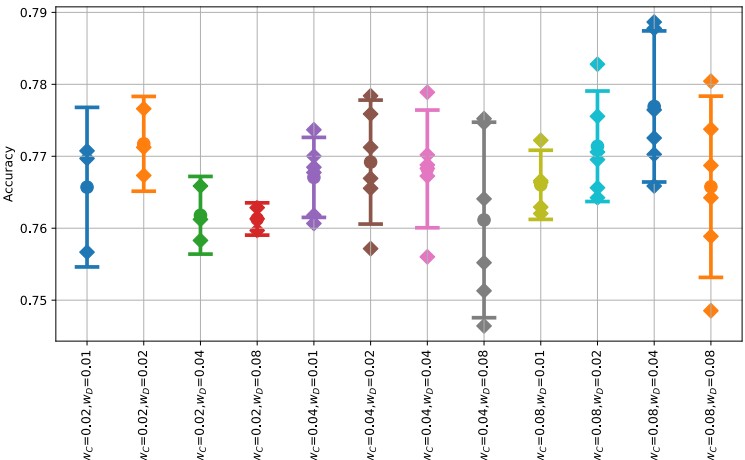

Figure 8: Specialized model accuracy for different values of $w_C$ and $w_D$.

### B.2.3 VAE Parameters.

In our experiments with in-context learning and text datasets we picked a simple uniform prior characterized by $\mathcal{K}_{i,j} = \nu\delta_{i,j} + (1-\nu)\mathcal{K}_{i,j}^{\mathrm{RBF}}$ with a sufficiently small $\nu$ and $\mathcal{K}_{i,j}^{\mathrm{RBF}} = \exp(-\|i - j\|^2/2\lambda^2)$. This choice is not optimal for many in-context learning tasks, where we expect less variation towards the end of the sequence, but it nevertheless allowed us to train high-performing models with interpretable context summaries.

Our VAE model was typically trained with $\nu = 0.03$ in the in-context learning setup and $0.1$ in text datasets. The characteristic auto-correlation size was chosen as $\lambda = 0.1$ ($10\%$ of the sequence length) and $\beta$ varied from $0.01$ to $10.0$. Additional experimental results and ablation studies can be found in Appendix D.4.

### B.3 In-Context Learning: Additional Details

In our experiments, we typically chose $n_{\mathrm{tasks}} = 4$ with $n_{\mathrm{ex}} = 4$ (with $n_{\mathrm{tasks}} = 1$ with $n_{\mathrm{ex}} = 8$ in some additional experiments outlined below). An example of a generated ASCII sequence before tokenization is:

```
154*709=+07058|648*011=+05920|526*187=+06230|893*495=+11997|#
122*395=-00273|827*301=+00526|216*082=+00134|399*879=-00480|#
913*075=+01063|748*228=+01204|508*205=+00918|186*523=+01232|#
349*703=+04547|343*849=+04785|868*591=+08994|124*356=+01828|#
```

All these lines concatenated together form a single sample. Here we put different tasks on different lines for clarity.

### B.4 Text Mixture Dataset: Additional Details

**8 different categories** "Mathematical identities" (0), "Real-time operating systems" (1), "Songs about nights" (2), "American abstract artists" (3), "Theoretical physics" (4), "State parks of Washington (state)" (5), "Film genres" (6) and "Three-ingredient cocktails" (7).

**Sample composed of 3 text excerpts.** Phrase composed of 3 different texts used in our experiments for verifying transitions of $\boldsymbol{y}^\ell$ (see Fig. 13(b)):

> The horned sungem (Heliactin bilophus) is a species of hummingbird native to much of central Brazil and parts of Bolivia and Suriname. It prefers open habitats such as savanna and grassland and readily occupies human-created habitats such as gardens. It recently expanded its range into southern Amazonas and Espirito Santo, probably as a result of deforestation; few other hummingbird species have recently expanded their range. The horned sungem is a small hummingbird with a long tail and a comparatively short, black bill. The sexes differ markedly in appearance, with males sporting two feather tufts ('horns') above the eyes that are shiny red, golden, and green. Linux was originally developed for personal computers based on the Intel x86 architecture, but has since been ported to more platforms than any other operating system. Because of the dominance of Linux-based Android on smartphones, Linux, including Android, has the largest installed base of all general-purpose operating systems as of May 2022. Linux is, as of March 2024, used by around 4 percent of desktop computers, the Chromebook, which runs the Linux kernel-based ChromeOS, dominates the US K–12 education market and represents nearly 20 percent of sub-$300 notebook sales in the US. Horse races vary widely in format, and many countries have developed their own particular traditions around the sport. Variations include restricting races to particular breeds, running over obstacles, running over different distances, running on different track surfaces, and running in different gaits. In some races, horses are assigned different weights to carry to reflect differences in ability, a process known as handicapping. Horse racing has a long and distinguished history and

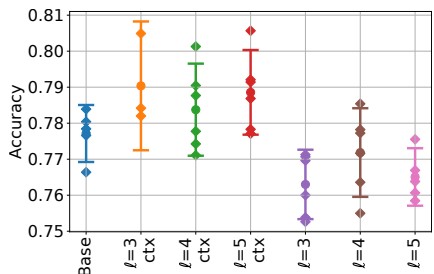

Figure 9: Average accuracies on the last two examples in different specialization runs for $\ell = 3, 4, 5$ (average and 3 times the standard error are also plotted): *with* previous examples in context (**ctx**) and *without* them.

> has been practiced in civilizations across the world since ancient times. Archae-
> ological records indicate that horse racing occurred in Ancient Greece, Ancient
> Rome, Babylon, Syria, Arabia, and Egypt.

## C  COVARIANCE FOR SIMPLE PARAMETER ESTIMATION PROBLEM

Consider an example of the sequence mean $\alpha$ estimation with a prior $\alpha \sim \mathcal{N}(0, 1)$ given a sequence of observations $\rho_s := \alpha + \epsilon\beta_s$ with $\beta_s$ sampled iid from $\mathcal{N}(0, 1)$.

The estimate of $\alpha$ after seeing observations $\{\rho_1, \ldots, \rho_s\}$ can be represented simply as

$$\hat{\alpha}_s := s^{-1} \sum_{i=1}^{s} \rho_i.$$

It is then easy to see that $\mu_s := \langle \hat{\alpha}_s \rangle = \langle \alpha \rangle = 0$ and the covariance matrix is given by:

$$\langle \hat{\alpha}_s \hat{\alpha}_t \rangle = \left\langle \left( \alpha + \frac{\epsilon}{s} \sum_{i \leq s} \beta_i \right) \left( \alpha + \frac{\epsilon}{t} \sum_{j \leq t} \beta_j \right) \right\rangle = \langle \alpha^2 \rangle + \frac{\epsilon^2}{st} \min(s, t).$$

As a result we see that the average $\hat{\alpha}_s$ across different sequences at every position is $0$ due to the symmetry of the problem and $\langle \alpha \rangle = 0$. On the other hand, computing the correlation of two estimates $\hat{\alpha}_s$ and $\hat{\alpha}_t$ in the same sequence, we will observe two contributions: (a) $\langle \alpha^2 \rangle$ contribution due to the fact that they share the same underlying realization of $\alpha$ and (b) the second term representing the decay of correlations due to the noise $\beta_s$ as we average over many elements and $s, t \to \infty$.

## D  ADDITIONAL EXPERIMENTAL RESULTS

### D.1  IN-CONTEXT LEARNING DATASET RESULTS

In addition to our experiments with $n_{\text{tasks}} = 4$ and $n_{\text{ex}} = 4$, we also conducted experiments using a single-task dataset ($n_{\text{tasks}} = 1$) with $n_{\text{ex}} = 8$ examples. The plot of the dot-product $\boldsymbol{n}_i \cdot \boldsymbol{n}_j$ (see Fig. 10(b)) can again be seen to reflect a gradual convergence of $\boldsymbol{y}^\ell$ as more and more examples are being processed (see Fig. 10(a)). Here we used a larger baseline model with 8 layers instead of 6, which reached the top accuracy of 82.7%. We then verified that multiple models trained with element-wise regularization, $\dim \boldsymbol{y} = 32$, $\ell$ being 4 or 5, softmax-based rank-4 matrix generator, our auxiliary loss and augmentation methods were able to achieve accuracies in the range 82.6% to 82.8% while using a *frozen* value of $\boldsymbol{y}^\ell$ obtained using several samples from the same task.

### D.2  LINEAR REGRESSION RESULTS

In our experiments with linear regression, we first analyzed models trained with both the cross-entropy and auxiliary losses ($\eta = 0.5$). Our base experiments were conducted with an 8-layer CGT, $\ell = 7$, $\dim \boldsymbol{x} = 128$, $\dim \boldsymbol{y} = 64$, rank $r = 4$ and the number of templates $M = 8$. In Figure 4 we

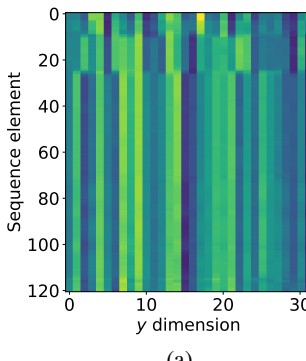
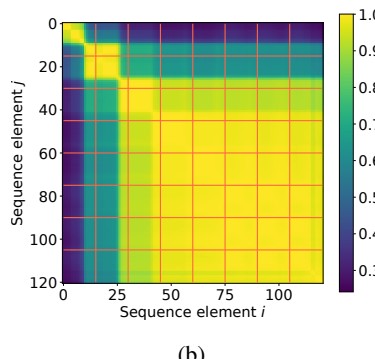

$$(a) \qquad\qquad (b)$$

Figure 10: **(a)** A typical dependence on $y^\ell$ on the sequence index for a synthetic in-context learning task with 8 examples; **(b)** dot product $n_i \cdot n_j$ for normalized $y^\ell$ embeddings at two different locations for this synthetic dataset.

show the evolution of a typical $L_2$ error between the base groundtruth value $Ux + b$ and the model prediction at token $x$ after seeing a given number of samples. As expected, the accuracy of the model prediction improves with the number of samples it observes in the sequence. The "baseline" curve illustrates performance of a conventional Transformer with $\dim x = 128$, while the "dynamic" plot is obtained with CGT model with token-to-token varying $y^\ell$. The same plot also shows similar curves for specialized models obtained by freezing $y^\ell$ after seeing 5, 10 and 20 examples. Figure 4 illustrates the fact that the accuracy of specialized models improves as we increase the number of samples employed for computing $y^\ell$. Horizontal dotted lines show the corresponding accuracy of the baseline model with the corresponding number of examples.

Studying the model behavior, we also conducted an ablation study varying different system parameters. We discovered that the rank of the generated low-rank matrices and the number of templates had virtually no effect on the model performance. However, the choice of the layer $\ell$ played a very large role. In Figure 11, we show the specialized model error curves obtained for different values of $\ell$ and the context size (used for computing $y^\ell$) of 20. We can see that the accuracy of the specialized model on the first $10 - 20$ examples is improving for higher $\ell$. In other words, the model benefits from using more layers for computing $y^\ell$. At the same time, the number of layers that $y^\ell$ modulates does not appear to be as critical.

We also studied the dependence of CGT performance on other parameters including $\dim x$ and $\dim y$. Increasing $\dim y$ improved specialized model performance immediately, even on small sequences. On the other hand, as shown in Figure 12, $\dim x$ proved to be critical for specialized model performance over long sequences (many additional samples on top of the task information communicated via frozen $y^\ell$).

### D.3 TEXT MIXTURE RESULTS

**Token probabilities.** We conducted additional experiments with specialized language models obtained by freezing $y^\ell$ value to a constant throughout the sequence. Specifically, we verified that replacing $y^\ell$ for one sequence with $y^\ell$ values from a different sequence has an expected impact on output token likelihoods. For example, by using $y^\ell$ from a "Theoretical Physics" page on a text from "American abstract artists" category, we observe that among top 500 tokens, the logits of "engine", "theory", "mechanics", "science", "condit", "chem", "physics" and other similar tokens, increased the most on average.

**Effect of regularization on $y^\ell$.** One way of looking at the effect of element-wise regularization on the representation $y^\ell$ is to study it's token-to-token change and at t-SNE plots of averaged $y^\ell$ for `wikipedia` articles from different topics. We see that adding element-wise regularization with $w_D = 0.04 = 2w_C$ leads to a much better clustering of representation $y^\ell$.

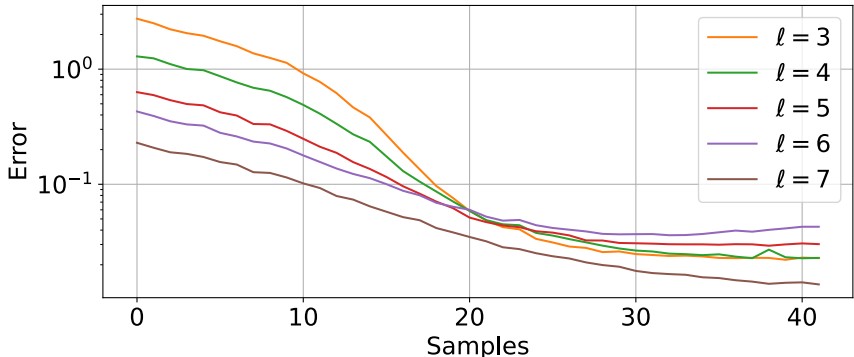

Figure 11: Average $L_2$ loss computed at a given sample index. We compare specialized models obtained after seeing 20 samples for 5 different values of $\ell \in [3, 7]$. Model behavior generally deteriorates towards the end of the sequence (for a large number of examples). Some models diverge after we observe more than $44 = 64 - 20$ samples, which is due to the fact that the model was trained with 64 samples in total and not all models generalize beyond sequence lengths seen during training.

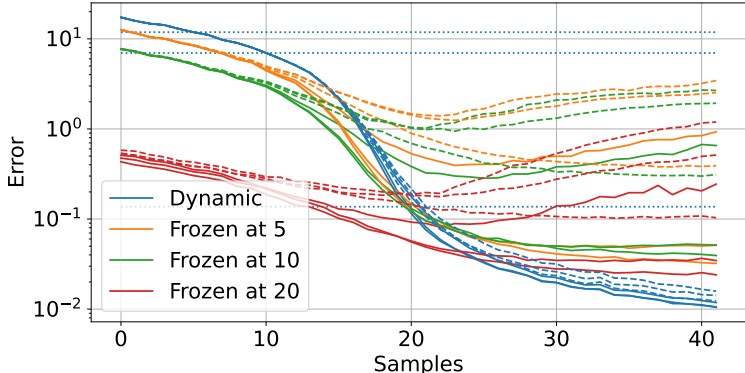

Figure 12: Comparison of $L_2$ errors for CGT model on the linear regression dataset with $\dim \boldsymbol{y} = 64$ and: (a) $\dim \boldsymbol{x} = 64$ (dashed), (b) $\dim \boldsymbol{x} = 128$ (solid). The plot shows 3 separate runs in both cases. One experiment with $\dim \boldsymbol{x} = 128$ shows degradation of performance for longer sequences.

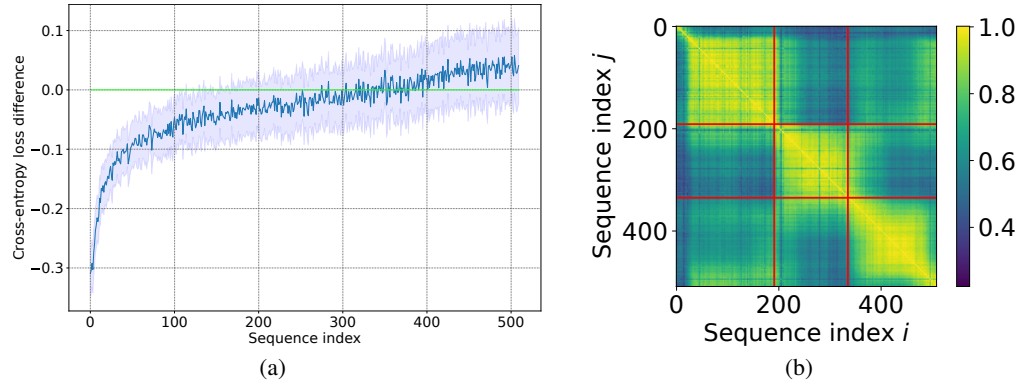

Figure 13: **(a)** average difference (on the second part of the document) between cross-entropies of a specialized model with $\boldsymbol{y}^\ell$ pre-computed on the first part and a baseline language model; **(b)** dot-product plot $\boldsymbol{n}_i \cdot \boldsymbol{n}_j$ for a combination of 3 different text excerpts described in Appendix B.4 (with boundaries shown).

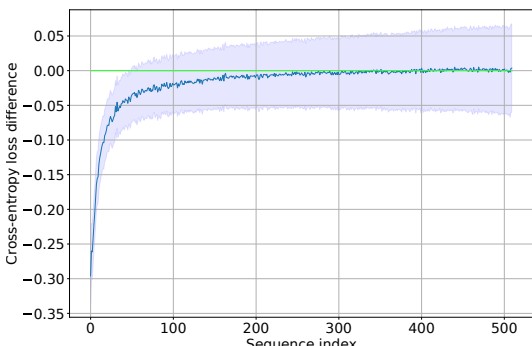

Figure 14: Average difference between the cross-entropy losses of the "informed" and "uninformed" (non-specialized) models. The informed model is generally better across the entire sequence (the difference is below zero). The informed model used dynamic value of $\boldsymbol{y}^{\ell}$ initialized with $(\boldsymbol{y}^{\ell})^{\text{init}}$ computed at the end of the first part and then maintained with a moving average with the rate $\gamma = 1/300$. In other words, we used $(\boldsymbol{y}^{\ell})_i^{\text{used}} = (1 - \gamma)(\boldsymbol{y}^{\ell})_{i-1}^{\text{used}} + \gamma(\boldsymbol{y}^{\ell})_i^{\text{computed}}$ with $(\boldsymbol{y}^{\ell})_0^{\text{used}} = (\boldsymbol{y}^{\ell})^{\text{init}}$. Maintaining this moving average allowed us to utilize information about the topic of the first part of the text without freezing $\boldsymbol{y}^{\ell}$ throughout the entire sequence. The uninformed model maintained a dynamic computed $\boldsymbol{y}^{\ell}$ without any direct or indirect access to the first part of the text.

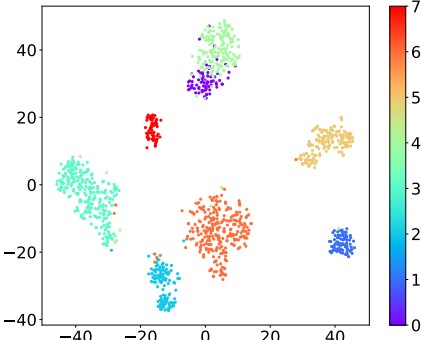

Figure 15: t-SNE plot for pages from $8$ `wikipedia` categories using a model trained on individual `c4` articles instead of pairs of randomly joined samples. This plot shows a much better separation between different categories, which is probably due to this test distribution being closer to the training set distribution (where each sample was generally touching a single topic).

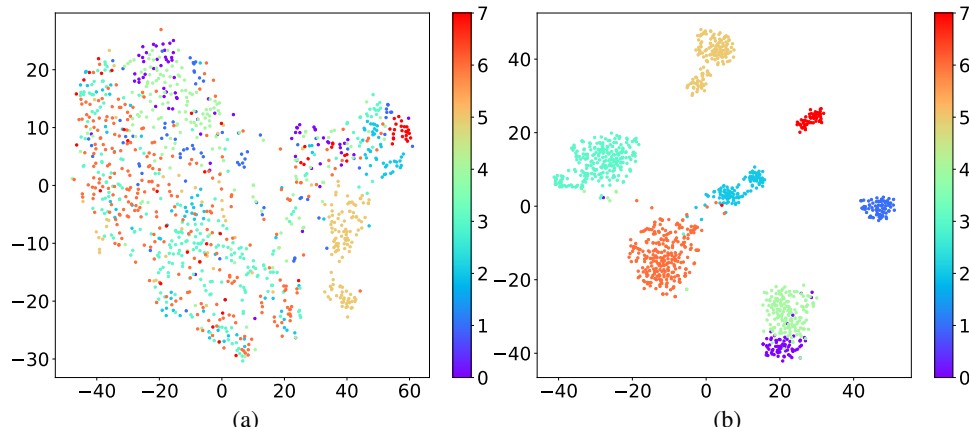

(a)                                              (b)

Figure 16: t-SNE plot for pages from $8$ `wikipedia` categories using a model trained on $2$ merged `c4` excerpts: (a) model without regularization; (b) model with element-wise regularization. The embeddings are obtained by averaging 16 sequential values of $\boldsymbol{y}^\ell$ at the end of the text. Considering instantaneous values of $\boldsymbol{y}^\ell$ results in similar plots.

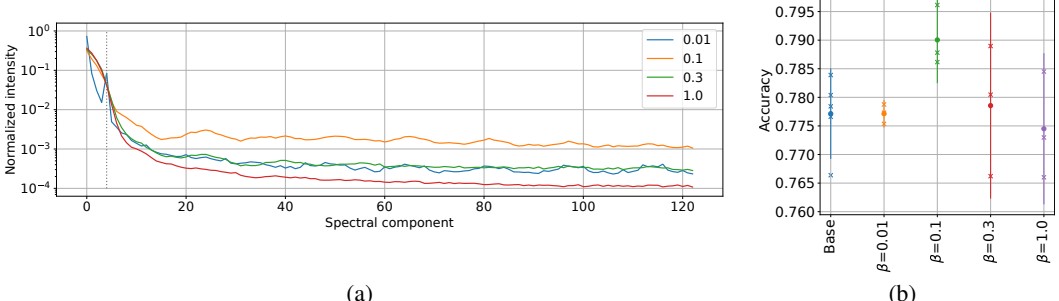

(a)                                              (b)

Figure 17: **(a)** Normalized averaged intensity $\langle |f_k|^2 \rangle$ of discrete Fourier transform spectra $f_k$ of all $\boldsymbol{y}^\ell$ components for VAEs with $\beta$ equal to 0.01, 0.1, 0.3 and 1.0. The averaging is performed over all components of $\boldsymbol{y}^\ell$ and over $256$ samples. The averaged intensity is then normalized to $1$ for each experiment for comparison. The model with $\beta = 0.01$ can be seen to have a peak around the $4^{\text{th}}$ harmonic (4 tasks). As $\beta$ increases, the spectrum smooths and higher harmonics disappear; **(b)** Model accuracies measured for the last 2 examples in VAE models with different $\beta$ values (showing individual accuracies, means and $3\sigma$).

### D.4 VAE RESULTS

The effect of varying $\beta$ in our VAE experiments with the in-context few-shot learning dataset are shown in Figure 17. We trained multiple models with different values of $\beta$ and observed that the model with $\beta = 0.01$ and hence virtually non-existent KL divergence term exhibited strong periodicity (on task boundaries), but as we increased $\beta$, model activations $\boldsymbol{y}^\ell$ became smoother (see Fig. 17(a)). Also, while for smaller $\beta$, the model tended to encode some task information in rapidly changing activation components, this behavior almost vanished at higher values of $\beta$ and model activations became a good predictor of the task multipliers $a$ and $b$. The effect of $\beta$ on model accuracy was also unsurprising in that strong regularization with higher values of $\beta$ appeared to hurt model performance (see Fig. 17(b)) suggesting that there might be a minor conflict between learning maximally useful representations $\boldsymbol{y}^\ell$ and these representations adhering perfectly to our desired prior.

Additional VAE results with `c4` dataset and varying values of $\beta$ are presented in Fig. 18, 19 and 20. First we show the dot-product $\boldsymbol{n}_i \cdot \boldsymbol{n}_j$ on a mixture of 3 distinct texts described in Appendix B.4

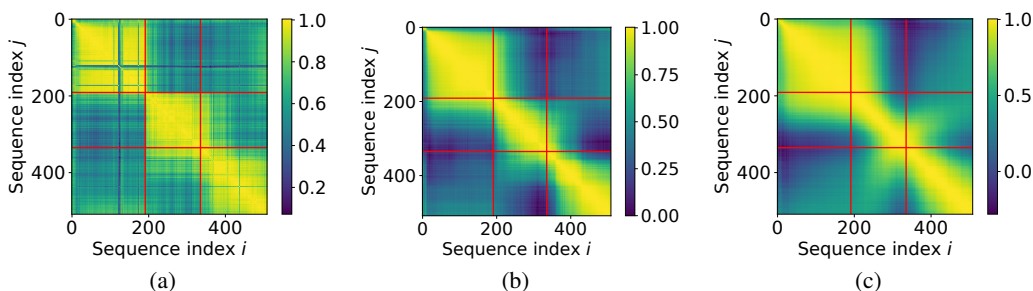

Figure 18: Dot product $\boldsymbol{n}_i \cdot \boldsymbol{n}_j$ plot computed for 3 different VAE models trained on c4 and evaluated on a mixture of 3 distinct texts (see Appendix B.4): (a) $\beta = 1$, (b) $\beta = 3$, (c) $\beta = 10$.

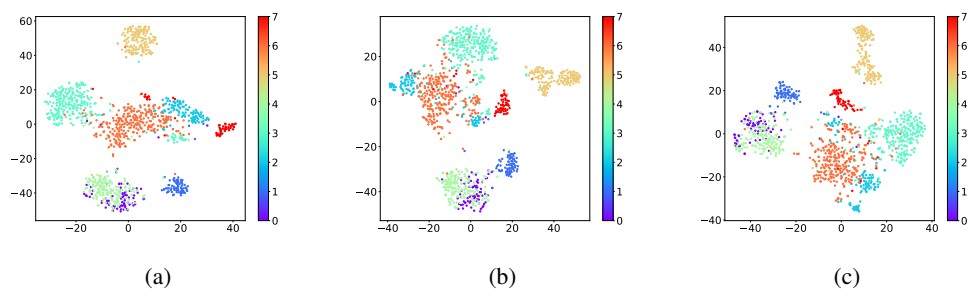

Figure 19: t-SNE plots for 3 different VAE models trained on c4 and evaluated on wikipedia pages from 8 distinct categories: (a) $\beta = 1$, (b) $\beta = 3$, (c) $\beta = 10$.

for different values of $\beta$ (Fig. 18). We then illustrate t-SNE plots of learned features on 8 distinct wikipedia categories (Fig. 19). Finally, in Fig. 20, we show traces of $\boldsymbol{y}^\ell$ activations on a mixture of 3 texts. It can be seen that increasing $\beta$ makes learned slow activations much smoother.

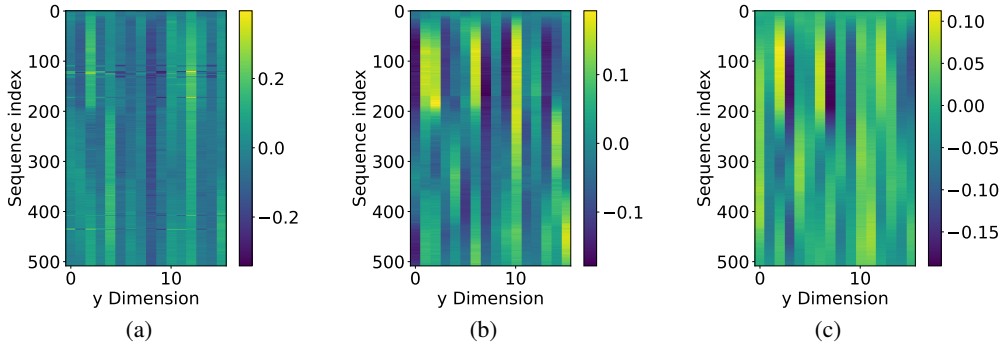

Figure 20: Context representation $\boldsymbol{y}^\ell$ evolution along the sequence for 3 different VAE models trained on c4 and evaluated on a mixture of 3 distinct texts (see Appendix B.4): (a) $\beta = 1$, (b) $\beta = 3$, (c) $\beta = 10$.

