# OpenReview forum: "Contextually Guided Transformers via Low-Rank Adaptation"
_ICLR.cc/2025/Conference — Submitted to ICLR 2025_

### Official Review · Reviewer_gek6 · 2024-11-02

**Soundness:** 3
**Presentation:** 3
**Contribution:** 2
**Rating:** 3
**Confidence:** 4

**Summary:**

The paper presents a modification to the Transformer architecture that integrates prompt context into the model's weights and activations. This integration enables the model to perform in-context learning without relying on preceding examples. To enhance the learned weights, the paper introduces an auxiliary loss and regularization techniques, resulting in smoother and more consistent results. To validate the effectiveness and interpretability of this approach, the paper conducts a series of experiments.

**Strengths:**

1. The presentation of the method is clear, providing a comprehensive introduction to its intricacies and implementation.
2. The theoretical support provided is convincing, extending across multiple research fields to demonstrate its validity in diverse contexts.
3. The experiments conducted are substantial and demonstrate the interpretability aspect of the method through multiple tests and analyses.

**Weaknesses:**

1. The proposed method needs to train from scratch for each distinct task, which is not a efficient way.
2. The quantitative results presented in Table 1 exhibit significant interval overlap, thereby reducing their statistical significance.
3. The applicability of the method remains uncertain due to its development on synthetic datasets and the necessity of training from scratch, as opposed to utilizing real-world datasets and adapting pretrained models.

**Questions:**

1.	If the model necessitates training from scratch for each distinct task, it would become prohibitively expensive. Consequently, one might consider training a LoRA module as a more efficient alternative. Please provide further elaboration on the motivations.
2.	How much time can be saved by eliminating the examples, compared with the time required to train the model?

---

> ### Author Response · Authors · 2024-11-20
>
> We greatly appreciate the reviewer's comments and suggestions, and their positive view of our presentation, theoretical foundation, and experimental design. We will address the weaknesses and concerns raised, but first want to emphasize a potential misunderstanding of our method. As we will clarify, our model does not require retraining for individual tasks; it only needs to be trained once within a particular domain. Specializing to new incoming tasks is handled by a single forward pass of the model with the task prompt, eliminating the need for in-context learning or prompt tuning with multiple examples. Moreover, as the reviewer suggests, leveraging pre-trained models is indeed feasible and advantageous.
>
> **W1.** _The proposed method needs to train from scratch for each distinct task, which is not an efficient way._
>
> Even when training from scratch, the model typically requires only a single pre-training phase. Once pre-trained, it can be applied to any new task without further retraining. This is achieved by simply feeding the model a prompt describing the task (e.g., "Translate from language A to language B" or "Perform the task demonstrated in the following 10 examples...") and extracting the final value of $y$ at the end of the prompt. This $y$, obtained through a single forward pass, is then used to generate a new Transformer model specialized for the task using straightforward algebraic expressions, without any training or fine-tuning.
>
> Furthermore, we can indeed utilize pre-trained models effectively. The pre-trained model can be frozen (or LoRA-tuned with our final objective) and used for computing $x$ activations. We then introduce new, potentially lightweight Transformer heads operating on $y$, trained alongside the frozen (or tuned) $x$ heads on a smaller dataset. These additional heads learn to aggregate task information from the prompt and adjust the weights of the underlying pre-trained model (above some layer) for better specialization to the given context. We are actively working to extend our experimental framework to support 1B- to 8B-parameter pre-trained LLMs. This is a key focus of our ongoing research.
>
> **W2.** _The quantitative results presented in Table 1 exhibit significant interval overlap…_
>
> While statistical fluctuations may limit the precision of our ablation studies and parameter analysis, they still demonstrate our core finding: it is possible to learn a mapping from a task-defining prompt to model weight updates, enabling the updated model to perform the task without the prompt nearly as well as the original model with the prompt.
>
> **W3.** _The applicability of the method remains uncertain due to its development on synthetic datasets…_
>
> Although most experiments used synthetic datasets, our language experiments utilized the C4 dataset, a widely used, large-scale corpus of real text data. Our model successfully specialized to previously seen context, as shown in Figure 6a.  While encouraging, we acknowledge these results are preliminary and further empirical investigation is needed to fully assess the practical capabilities of our method.
>
> **Q1.** _… one might consider training a LoRA module as a more efficient alternative. Please provide further elaboration on the motivations._
>
> We've already touched on this in our response to **W1**, but to reiterate: our goal is to pre-train a single, comprehensive model on a massive dataset. This initial pre-training is computationally expensive, but it allows the model to adapt to new tasks incredibly efficiently using prompts.
>
> Effectively, we do the heavy lifting upfront. Then, for any new task, we simply provide the model with a specific prompt, and it generates a specialized $x$-only version of itself. This process requires minimal computation – just a single forward pass through the model. Compared to traditional fine-tuning methods like LoRA, our approach offers significant cost savings, especially when scaling to a large number of tasks.
>
> **Q2.** _How much time can be saved by eliminating the examples, compared with the time required to train the model?_
>
> Our model's ability to handle any prompt-defined task after a single training phase leads to significant computational savings. Generating a new sequence of length $s$ from a prompt of length $r$ with an $l$-layer model typically involves $rsl$ cross-attention operations. With static prompts, this cost is incurred for every generated sample. Our approach, while slightly more expensive than running the original model (typically by a factor $<2$), eliminates the context and its associated cross-attention computation by integrating it into the model weights. This makes our method advantageous even with minimal prompt reuse, and the savings become substantial with frequent prompt reuse.

---

> ### Comment · Reviewer_gek6 · 2024-11-26
>
> Thanks for your response.  however, I am not satisfied with your response, especially for why not conducting experiments on real tasks with real datasets. So, I don't change my score.

---

> > ### Author Response · Authors · 2024-11-26
> >
> > Thank you for your feedback! We understand you're not satisfied with our dataset choices. We're curious if you could advise us on specific examples of real-world datasets that would be more suitable for our purposes. We're also interested in your perspective on the C4 dataset we experimented with. (C4 is a massive, cleaned version of Common Crawl, containing approximately 364 million real-world text sequences.)

---

### Official Review · Reviewer_Yv8f · 2024-11-03

**Soundness:** 2
**Presentation:** 1
**Contribution:** 2
**Rating:** 3
**Confidence:** 3

**Summary:**

This paper proposes an alternative to using explicit prompts for specializing LLMs. The authors introduce a method called Contextually Guided Transformer. This method processes the context text with a lighter-weight model and generates weight adaptation for the full model. Furthermore, to improve computation efficiency, the authors introduced auxiliary loss, smoothness prior, and VAE to allow the model to keep the weight adaptation fixed in the full model. The authors perform experiments on two synthetic toy tasks and one language modeling task and support the results with various analyses.

**Strengths:**

* This paper proposed an innovative solution to an interesting problem, leveraging thoughts from various research directions, such as adapter/lora, low-rank decomposition, VAE, and hyper tuning.

* The authors have performed a plethora of different experiments and analyses. These results help reveal how the method works.

**Weaknesses:**

* The datasets used in this work have limited information. But, to answer whether we can encode context into weight adaptation, we need tasks like reading comprehension. More specifically, in the synthetic dataset, the model only needs to encode two numbers from the context; in the linear regression dataset, the model needs to encode 16 numbers; finally, in language modeling, the benefit can be mostly captured by simply encoding the topic (there are eight in total), as partially suggested in the t-SNE analysis.

* Writing is relatively unclear. There are multiple places where the method or experiment setting is ambiguous. (see the questions for details)

* Limited successes in experiments. Although there are encouraging results in the toy datasets, in language modeling, the effect is not (potentially due to undesirable experiment setting rather than shortcomings of the method).

* Although saving the cost of explicit prompting is a main objective, this paper does not compare costs either theoretically or empirically. Ideally, we should compare (i) not fixed (ii) fixed (iii) baseline (iv) baseline with cached kv from context.

**Questions:**

* I need some help understanding the experiment setting. In the three datasets, which part of the text is treated as y, and which part is treated as x? Could you provide some examples in the rebuttal?

* I didn't manage to understand the model's architecture. How is the representation of x and y produced from the GPT-2 model? When we change the x and y dimensions, does that also affect the model's hidden dim? And how do we use different x and y

* In the language modeling experiment, I don't understand the reason for cutting and concatenating two text samples. I would expect using a single example as the more logical choice for measuring the model's ability to use context. If the text comprises two independent and incomplete sentences, the model benefits less from short-term context (it can break up at any point) or long-term context (the early tokens may belong to an irrelevant example).

* Line 337 whcih -> which

---

> ### Author Response · Authors · 2024-11-20
>
> We would like to thank the reviewer for their careful consideration of our publication and their comments.
>
> **W1.** _The datasets used in this work have limited information… we need tasks like reading comprehension… in language modeling, the benefit can be mostly captured by simply encoding the topic (there are eight in total)_
>
> We certainly agree that an in-depth study of the practical merit of our approach would require more careful experimentation with reading comprehension, textual many-shot in-context learning and other similar practical datasets. This evaluation is the subject of our on-going and future work and we hope to present a more in-depth evaluation in the near future. It is also worth pointing out  that some of our experiments reported in the paper have been conducted with the C4 dataset (see Figure 6a), which is a very large and widely used natural language dataset. While we do use only 8 wikipedia topics to visualize the properties of the learned embeddings, the original C4 dataset does not have a restricted set of topics and therefore, our experiments with this dataset do already exhibit a form of real reading comprehension where any factual knowledge extractable from the first half of the text needs to be utilized to generate the rest of the sequence.
>
> **W2.** _Writing is relatively unclear. There are multiple places where the method or experiment setting is ambiguous._
>
> We will revise the text to provide more intuitive and understandable descriptions of the core model components. We will also include additional explicit algorithms to make sure that all of these components are presented without ambiguity.
>
> **W3.** _Limited successes in experiments. Although there are encouraging results in the toy datasets, in language modeling, the effect is not (potentially due to undesirable experiment setting rather than shortcomings of the method)._
>
> Our current experimental results are at least partially limited by the size of the employed GPT-2 models. The model capabilities and the modulation of the model weights are expected to be more substantial once we reach the model size of 1B. We are presently preparing to make it possible to use pre-trained 1B-8B Language Models and we are hoping to evaluate the capabilities of our technique in this more practical setup soon.
>
> Overall, the primary goal of this publication was to propose a novel approach and draw the attention of the community to a promise of combining the TTT paradigm (learning at inference time) and hypernetwork-inspired weight modulation mechanism. We wanted to establish its core properties on simple yet hopefully insightful model tasks while making a much more in-depth practical evaluation a subject of the followup work.
>
> **W4.** _Although saving the cost of explicit prompting is a main objective, this paper does not compare costs either theoretically or empirically. Ideally, we should compare (i) not fixed (ii) fixed (iii) baseline (iv) baseline with cached kv from context._
>
> This is another excellent point and we will update the publication to include a more in-depth discussion of the computational complexity in these different scenarios.
>
> The advantage of moving the context into the weights can be clearly seen in the asymptotic case of using a very large amount of tasks (specified via prompts) and evaluations per task. If both the number of tasks and number of evaluations are very large, we can disregard the cost of training our model (either from scratch or based on tuning a pre-trained model). Furthermore, even though we use a simple forward pass through our model on a prompt to generate a specialized model, this cost will eventually be completely offset as the number of evaluations per prompt grows. While the model using the prompt in-context will always use an additional compute to cross-attend to a prompt KV cache, our specialized model will not have this associated extra cost (we embed information from the prompt into the model weights and then drop the prompt altogether). The proposed approach therefore reaches the lowest possible complexity in the limit of infinitely many inferences per task (effectively specializing the model to perform a particular task once and then running this specialized model on a variety of inputs).

---

> ### Author Response · Authors · 2024-11-20
>
> **Q1 and Q2**. Questions about the model.
>
> We are sorry that our explanations were not entirely clear and will try to clarify these points here.
>
> **[Figure](https://i.postimg.cc/6BxfjS4w/Model-Illustration.png)**. (left) Conventional Transformer model with 3 input tokens with dotted lines showing cross-attention to prior tokens and (right) our architecture with 2 input tokens and $\ell=2$. All newly introduced elements are shown in blue; both $x$ and $y$ activations are computed for each input token, but $x$ have only the visibility of other $x$ activations; residual low-rank linear operators $T^{\kappa}$ depend on $y^{\ell}$ and augment $x$ at each token in all layers above $\ell$.
>
> In our base model, the entire input sequence is used to generate both $x$ and $y$ activations. The $x$ activations can be thought of as activations of a conventional base Transformer model (think of this as a base GPT-2) processing the entire input sequence (for instance, we could choose the $x$ embedding size of 256 with 8 heads each having 32 dimensions). On top of that, in between layers 1 and $\ell$ (some pre-selected layer), we also introduce additional Transformer heads responsible for producing new $y$ activations (for instance, we could add extra 4 heads with the same dimension 32 thus making the dimension of $y$ equal to 128), again for every single input token. These new heads are introduced in such a way that they do not impact the computation done in $x$ anywhere up to the layer $\ell$. However, when computing $y$ itself, we allow it to observe $x$ components (both in MLP and in self-attention layers).
>
> After layer $\ell$, the $y_\ell$ activations are no longer updated and are only used for modulating the computation on $x$ in all subsequent layers. These activations $y_\ell$ are used to produce a variety of linear operators $T^{\kappa}$ that update activations $x$ in between consecutive layers after $\ell$. Notice that this happens for every incoming token and could be viewed as a simple computational modification of a conventional Transformer per-token information flow. The main difference is that now it loses a conventional layer-wise computational structure.
>
> This is the description of the base architecture, which can be viewed as just a modified sequence model. However, we now aim to make $y$ not just another per-token embedding, but endow it with a special function and meaning by introducing an additional auxiliary loss. Specifically, we want to be able to “freeze” $y$ at any token and then use linear operators $T^{\kappa}$ generated for this frozen value for the remainder of the sequence without the preceding part being in context. In doing so, we effectively make $y$ aggregate information about the prior context and use this information to modify the sequence model on the second “half” of the sequence. This can be viewed as another instance of TTT (train at test time), where $y$ is effectively learning something about the sequence while the sequence is being processed token-by-token.
>
> We compute the auxiliary loss by randomly cutting each sequence into two parts and effectively dropping the first half while only retaining (and freezing) the $y_\ell$ activations computed on it (see Figure 1b). For example, in a typical C4 sequence, we would just: (a) randomly cut an entire text at some token, (b) use the first half to compute $y_\ell$ as if the first half was the “prompt” specifying the task, or providing some important information, (c) we then generate weight updates for this value of $y_\ell$ on the remainder of the sequence and effectively use this generated and frozen model on the second half. If our model is successfully learning to convert the prompt (first half) into the weight update (with the updated model applied to the second half), then the cross-entropy loss should go down in the second half, where the model has now absorbed some relevant information about the first half via updated weights.

---

> ### Author Response · Authors · 2024-11-20
>
> **Q3.** _In the language modeling experiment, I don't understand the reason for cutting and concatenating two text samples._
>
> This is a good point that we need to clarify. It is indeed true that using a single contiguous sequence would be more convenient in that it would allow us to treat the second half (with the sequence split at some point) as a logical continuation of the first half. We used this more complex two-excerpt setup simply to demonstrate that our approach can discover useful context representations even when the sequence may contain multiple completely unrelated pieces of information. In our physical world, the sensory data is frequently overwhelmed with quickly changing contexts and it was important for us to design and demonstrate a technique that could discover a representation that could be useful at least locally (via self-modulation), while not being confused by the changing environment. You can think of it as developing a flexible short-term memory that can be attained and improved upon and then overwritten when the context shifts. This was viewed as additional evidence of the robustness of this technique.

---

> > ### Comment · Reviewer_Yv8f · 2024-11-29
> >
> > Thank you for your explanation. I'm not convinced about the experiment setting, but I appreciate the clarification and novelty of the idea.

---

> > > ### Author Response · Authors · 2024-12-03
> > >
> > > Thank you for your feedback and suggestions! While we would be interested to learn more about the types of experiments that you feel could be more convincing, we will take this general feedback into consideration and it will undoubtedly help us refine the publication.

---

### Official Review · Reviewer_ns7G · 2024-11-11

**Soundness:** 3
**Presentation:** 3
**Contribution:** 3
**Rating:** 6
**Confidence:** 4

**Summary:**

This paper proposes a modified Transformer model called the Contextually Guided Transformer (CGT). To achieve self-modulation, it uses a context summary embedding at each layer to update subsequent layers’ weights by balancing the original task cross-entropy loss with a context auxiliary loss. CGT shows good performance in specialized tasks, especially in initial context-critical tasks, and is more efficient than current methods, which often rely on repeated attention-guiding prompts or an additional model to guide the main Transformer’s weight updates. To improve interpretability of the context representation, the paper also introduces element-wise regularization by viewing the Transformer as a Variational Autoencoder (VAE), ensuring the context summary embedding evolves gradually.

**Strengths:**

1.This paper introduces an innovative architecture that enhances Transformer behavior on more specialized tasks. Instead of using repetitive prompts or training an additional model—which increases computation cost and complexity—it introduces self-specialization by exploring the different potential between global context and local context of embeddings.
2. They introduce a simple regularization scheme by viewing the transformer as VAE.
3.The design of the loss function and regularization function is elegant and well-explained, contributing to clearer demonstrations and results.

**Weaknesses:**

1.The experiments only compare CGT with a traditional autoregressive causal Transformer as a baseline across limited datasets (types of tasks). The results would be more persuasive if other modified Transformers for specialized purposes were included in the comparison.
2.The paper finds that consecutive tokens in the context summary embedding do not behave smoothly, contrary to expectations. An explanation of why this phenomenon occurs would improve the paper.

**Questions:**

See weakness.

---

> ### Author Response · Authors · 2024-11-20
>
> We would like to thank the reviewer for their comments, suggestions and their positive view of our proposed technique. In the following, we address weaknesses pointed out by the reviewer:
>
> **W1.** _The results would be more persuasive if other modified Transformers for specialized purposes were included in the comparison._
>
> While our current work focuses on autoregressive causal Transformers, we recognize the potential of the proposed technique to extend to other architectures. We are particularly interested in exploring its application to models that combine recurrent layers (e.g., state space models) with self-attention layers. One promising avenue is to utilize linear recurrent layers for maintaining a dynamic task summary ($y$), while employing local attention layers operating on $x$ for localized sequence processing. However, we believe a thorough investigation of these alternative architectures requires extensive exploration. Since our goal in this publication was to introduce our core approach, such in-depth analysis of other architectures felt to be outside the scope of this initial publication. Nevertheless, we consider this a compelling direction for future research.
>
> **W2.** _The paper finds that consecutive tokens in the context summary embedding do not behave smoothly, contrary to expectations. An explanation of why this phenomenon occurs would improve the paper._
>
> We appreciate the reviewer highlighting this potentially counter-intuitive result. Following this feedback, we will provide further discussion in the main text to hopefully clarify this point a little. In our opinion, the core reason for this is that even with the auxiliary loss, the model is not directly incentivized to make $y$ behave smoothly.
>
> For example, imagine that produced $y$ activations contain some components that are pseudo-random and change rapidly token-to-token. If the following linear operators $T^{\kappa}$ simply ignore these components, then the final loss would not be impacted by them even with the auxiliary component. Informally speaking, while the optimization procedure running for a very long time, could eventually lead to all components of $y$ performing some useful computation, low incremental utility of some of these components could cause them to remain under-utilized and rapidly changing when trained for a shorter duration (essentially containing noise).
>
> In addition, our intuition suggests that it is also possible that local token-to-token computation could also be partially allocated to at least some of the components of $y$. While removing these local changes in $y$ would affect predictions when computing the auxiliary loss (since we effectively freeze $y$ to be constant), this additional local information and an associated computational capacity could still be utilized in the main loss when running on the original un-augmented sequence. Optimizing the sum of two losses (on the original and augmented sequences), we could thus still incentivize the model to partially utilize some of the $y$ components for local computation.
>
> These two hypothetical explanations could perhaps clarify why directly forcing $y$ to be smooth may improve generalization of the final model by removing the noise and overfitting in the components of $y$.

---

> > ### Comment · Reviewer_ns7G · 2024-11-22
> >
> > Thanks the authors for their feedback. It has solved my concerns.

---

> > > ### Author Response · Authors · 2024-12-03
> > >
> > > Thank you very much for your feedback and suggestions! They were incredibly helpful and will allow us to make significant improvements in the following iteration.

---

### Official Review · Reviewer_G5Ls · 2024-11-11

**Soundness:** 3
**Presentation:** 2
**Contribution:** 3
**Rating:** 5
**Confidence:** 4

**Summary:**

The Contextually Guided Transformer (CGT) paper introduces a novel Transformer model that incorporates context adaptation directly into its architecture. Unlike traditional Transformers that rely on prompts or fine-tuning to adapt to new tasks or contexts, CGT dynamically adjusts its processing based on a context summary that evolves as the model reads through the input sequence.

**Strengths:**

The CGT introduces a unique approach to embedding context into the Transformer’s architecture. By using y-components (context summaries) to dynamically modulate the model, CGT offers an innovative solution to integrate context without the need for explicit prompting at every layer.


The auxiliary loss is a thoughtful addition, penalizing large changes in the context summary to encourage smooth, interpretable, and stable evolution of the y-components. This loss term is well-conceived.


CGT holds promise for applications requiring efficient, context-specific adjustments to input sequences, potentially reducing the need for resource-intensive prompt processing in context-sensitive tasks.

**Weaknesses:**

The paper’s explanation of core concepts, especially around the auxiliary loss, low-rank transformation of y-components, and the role of x- and y-components, is highly technical but but lacks clarity. The separation of x- and y-components and the exact mechanism of context modulation are only partly clear after multiple readings and external clarifications.
In particular, sections describing the technical architecture, auxiliary loss, and low-rank transformations should be simplified, with more intuitive explanations.

The experiments in the paper are insufficient to support the claims made about CGT’s adaptability and efficiency. The limited range of benchmarks leaves questions regarding CGT’s real-world performance on common language tasks (e.g., machine translation, text classification, etc.).

Comparisons are mostly against standard Transformers, which do not offer context adaptation; comparisons with strong baselines in adaptive transformer architectures, such as [Dynamic Evaluation (DE)](https://proceedings.mlr.press/v80/krause18a.html) and other [Test-Time Training]() methods, are necessary to verify if CGT’s approach is truly advantageous.

The paper claims that CGT is a flexible, context-adaptive model, yet the provided experimental results do not provide convincing evidence of this. The model’s claimed efficiency and ability to improve over static Transformer architectures are not well-validated.
Moreover, the choice of synthetic tasks and limited application domains makes it difficult to gauge how well CGT would perform in more complex or real-world tasks. A more robust evaluation on diverse benchmarks would provide stronger empirical support for CGT’s effectiveness.

**Questions:**

Will the code be open-sourced for this work?

---

> ### Author Response · Authors · 2024-11-20
>
> We would like to thank the reviewer for their very careful and in-depth consideration of our publication and their comments. We are glad to receive an overall positive feedback to the core idea behind our technique. While we address the weaknesses highlighted by the reviewer below, we overall agree with all of the areas for improvement brought up in this review. We will do our best to update the paper and address the clarity issue.
>
> **W1.** _The paper’s explanation of core concepts … lacks clarity._
>
> Receiving feedback from all of the reviewers, we see a lack of full clarity being a common theme and will revise the text to provide more intuitive and understandable descriptions of the core model components. We will also include additional explicit algorithms to make sure that all of these components are presented without ambiguity.
>
> * The $x$ pathway processes information locally (like a standard Transformer), while the $y$ pathway accumulates global context from all preceding tokens. In layers above $\ell$, $y^\ell$ modulates the computations through learned transformations $T^k(y^\ell)$, effectively specializing the model for the following subsequence. Freezing $y^\ell$ creates a specialized model with fixed weights, operating solely on $x$.
> * The auxiliary loss trains the model to predict the sequence suffix ($t_{>s}$) based only on $y^l$ computed at a random split point $s$. This forces $y^\ell$ to become a compact summary of the prefix ($t_{<s}$) — all the information needed to process what follows. Specifically, at a random position $s$, $y^\ell$ is extracted from the full model operating on the entire sequence. A specialized model is then constructed using this fixed $y^\ell$, which is applied only to the suffix $t_{>s}$ and the loss is calculated on this portion.
> * The low-rank transformations $T^k(y^\ell)$ are crucial for two reasons: (1) _Efficiency_: They reduce the number of parameters compared to full-rank transformations, making context modulation computationally manageable. (2) _Interpretability_: They represent the context's influence as a linear combination of a small set of learned "template" transformations, making the modulation mechanism easier to understand. This low-rank structure may also offer a regularizing effect.
>
> We hope that these clarifications address the reviewer's concerns and provide a more intuitive understanding of our proposed method. We are happy to provide further details or examples if needed.
>
> **W2 and W4**. _The experiments in the paper are insufficient to support the claims made about CGT’s adaptability and efficiency. – … experimental results do not provide convincing evidence … makes it difficult to gauge how well CGT would perform in more complex or real-world tasks._
>
> We acknowledge the reviewer's concern regarding the scope of our experiments. The chosen tasks – synthetic in-context learning, linear regression, and text mixture modeling – were specifically designed to isolate and highlight CGT's ability to learn and utilize contextual information for specialized model generation. However, we agree that evaluating CGT on more complex real-world tasks is essential for a comprehensive assessment of its potential. Future work will focus on extending these experiments to more challenging domains, such as machine translation, question answering, and few-shot learning on more diverse datasets.
>
> Notice that using the proposed technique with a pre-trained model (for $x$ heads) is guaranteed to provide some level of context specialization by design. However, it remains to be seen whether generated specialized models operating without the context can match the performance of models with context on practical tasks. This evaluation of the practical merit of our method is currently ongoing and we will attempt to include additional evaluations in our paper.
>
> **W3.** _Comparisons are mostly against standard Transformers…; comparisons with strong baselines in adaptive transformer architectures … are necessary to verify if CGT’s approach is truly advantageous._
>
> We agree that a proper evaluation of our approach should ideally include such comparisons with other adaptive transformer architectures and we thank the reviewer for referring us to the DE paper (the second link for TTT seems to be unfortunately broken). While it remains to be seen whether the specialized models produced using our method are competitive to those obtained with alternative techniques, we see some potential benefits of our approach. For example, our method does not appear to rely on a proper hyper-parameter choice as much as perhaps DE does. It may also have a greater capacity to specialize to a particular data distribution. However, these are only speculations and we hope to learn more by comparing these techniques empirically.
>
> **Q.** _Will the code be open-sourced for this work?_
>
> We plan to open-source our code at the same time, or shortly after the publication.

---

### Comment · Area_Chair_zkPA · 2024-11-24

Dear Reviewers,

This is a gentle reminder that the authors have submitted their rebuttal, and the discussion period will conclude on November 26th AoE. To ensure a constructive and meaningful discussion, we kindly ask that you review the rebuttal as soon as possible and verify if your questions and comments have been adequately addressed.

We greatly appreciate your time, effort, and thoughtful contributions to this process.

Best regards,
AC

---

### Meta-Review · Area_Chair_zkPA · 2024-12-27

**Metareview:**

The paper proposes a new architecture mechanism to encode an arbitrary context into the weights of a transformer. This can reduce the context length, remove repetitiveness for different examples, and empirically leads to smoother representations. The authors conduct experiments on synthetic in-context learning and a limited set of language modeling experiments with a GPT-2 small model. The reviewers appreciated the novelty and motivation of the approach, but there were common concerns regarding its applicability to real-world tasks.

**Additional Comments On Reviewer Discussion:**

The reviewers appreciated the novelty and motivation of the approach, but there were common concerns regarding its applicability to real-world tasks. Specifically, it is not enough to train the model on a real-world text corpus but given the advances in LLMs, the bar for practical improvements is much higher and it is reasonable to expect empirical validation on downstream tasks (eg, instruction following, commonsense reasoning) which can benefit from the proposed approach. This is the main criticism of the work, and the basis for the final decision.

---

### Decision · Program_Chairs · 2025-01-22

Reject